# WritingBench: A Comprehensive Benchmark for Generative Writing

**Yuning Wu[1], Jiahao Mei[1,3], Ming Yan[1]\*, Chenliang Li[1], Shaopeng Lai[1], Yuran Ren[2],**
**Zijia Wang[2], Ji Zhang[1], Mengyue Wu[3], Qin Jin[2]\*, Fei Huang[1]**
[1]Alibaba Group, [2]Renmin University of China, [3]Shanghai Jiao Tong University
{yuningwu, ym119608}@alibaba-inc.com, qjin@ruc.edu.cn

## Abstract

Recent advancements in large language models (LLMs) have significantly enhanced text generation capabilities, yet evaluating their performance in generative writing remains a challenge. Existing benchmarks primarily focus on generic text generation or limited in writing tasks, failing to capture the diverse requirements of high-quality written contents across various domains. To bridge this gap, we present **WritingBench**, a comprehensive benchmark designed to evaluate LLMs across 6 core writing domains and 100 subdomains.We further propose a *query-dependent evaluation* framework that empowers LLMs to dynamically generate instance-specific assessment criteria. This framework is complemented by a fine-tuned critic model for criteria-aware scoring, enabling evaluations in style, format and length. The framework's validity is further demonstrated by its data curation capability, which enables a 7B-parameter model to outperform the performance of GPT-4o in writing. We open-source the benchmark, along with evaluation tools and modular framework components, to advance the development of LLMs in writing.

**GitHub Repo:** https://github.com/X-PLUG/WritingBench
**Leaderboard:** https://huggingface.co/spaces/WritingBench/WritingBench

## 1 Introduction

In recent years, LLMs [4, 8] have revolutionized text generation, demonstrating impressive performance across diverse applications, from generating creative content [19, 11] and assisting in education [27, 13] to enhancing professional workflows [25, 12]. However, generative writing, which requires high levels of coherence, creativity, logical reasoning, and stylistic precision, poses a unique challenge that existing evaluation methods fail to address adequately.

Current evaluation benchmarks for generative writing suffer from two major limitations: 1) Limited scope and diversity in task formulation; and 2) Inadequate evaluation metrics for complex writing tasks. First, there is a significant lack of specialized benchmarks that cover a broad range of writing scenarios. Most existing writing-oriented benchmarks are restricted to single domains, such as fictions [10, 7],

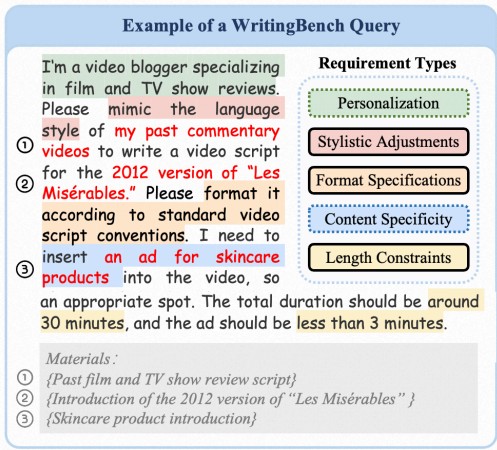

Figure 1: Query example with color-coded requirements. Red phrases correlate with gray-shaded writing support materials.

---

\*Corresponding authors

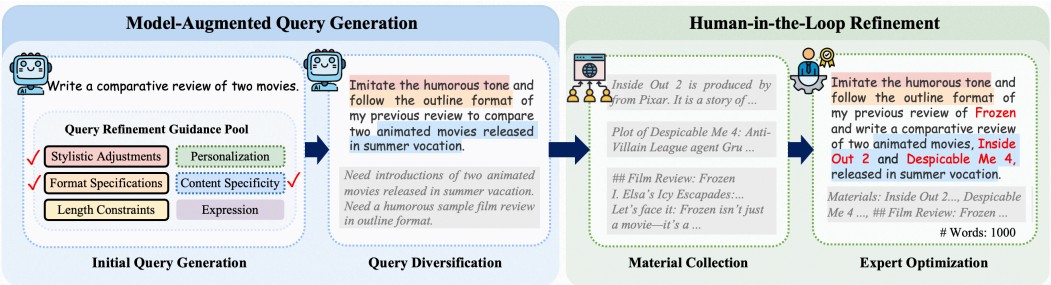

Figure 2: Construction pipeline of WritingBench queries. The refinement pool contains five writing requirements (three core competencies with black borders) and one expression type (purple). Checked strategies refine initial queries into multi-requirement prompts (color-coded text) with red phrases referencing gray materials. Implementation details in Section 3.1.

and their task formulations tend to be simplistic—often relying on single-sentence prompts [3] or a small set of instruction templates [21, 23]. Additionally, many benchmarks use homogeneous input materials [23, 10], limiting their ability to accommodate the complex and customized requirements inherent in real-world writing. As a result, they fail to capture the diversity and intricacies of practical writing tasks (see Figure 1). Second, current automatic evaluation metrics lack the robustness needed for a comprehensive and nuanced assessment of writing quality. While LLM-based evaluation methods show promise in capturing semantic meanings [25, 23, 3], they typically rely on a narrow set of predefined criteria (e.g., fluency and coherence). As LLMs continue to evolve with increasingly sophisticated writing capabilities, these static evaluation criteria and frameworks are inadequate for assessing the complex, multi-dimensional nature of writing, including creativity, argumentation strength, and domain-specific adherence.

To address these challenges, we introduce **WritingBench**, a comprehensive benchmark and robust framework for evaluating general-purpose writing. Our approach begins with a carefully designed secondary domain categorization, grounded in real-world writing needs. We develop a four-stage query construction pipeline (illustrated in Figure 2), where LLMs first generate and diversify writing queries, followed by human-driven material collection and optimization. This process ensures a diverse set of writing tasks with broad domain coverage, varied requirements, and integration of heterogeneous source of materials. To enable a more nuanced evaluation, we propose a *query-dependent evaluation* framework where LLMs dynamically generates five instance-specific criteria, which are then scored by a fine-tuned critic model. Finally, we integrate the framework to filter writing-specific data and train a small-scale model to verify its ability in identifying high-quality writing samples. Our primary contributions are as follows:

- We present **WritingBench**, an open-source writing benchmark comprising *1,000* queries across *6* primary domains and *100* subdomains, featuring *style*, *format* and *length* requirements. Writing-Bench supports extended-context generation with input ranging from tens to thousands of words, addressing real-world diversity. It facilitates systematic evaluation to identify improvement areas and highlights the potential of chain-of-thought (CoT) processes in creative tasks.

- We propose a *query-dependent evaluation* framework that integrates instance-specific criteria generation with a criteria-aware scoring model. It achieves 84% human alignment, significantly surpassing static-criteria baselines (67%, 58%). The effectiveness is further evidenced by its data curation capability-models trained with framework-filtered data outperform GPT-4o in writing.

- We publicly release WritingBench, including its evaluation protocols, criteria generation tools with an integrated critic model, and writing-enhanced models, to foster further research. Available at: `https://github.com/X-PLUG/WritingBench`.

## 2 Related Work

### 2.1 Writing Benchmarks

Existing writing benchmarks suffer from significant limitations in domain coverage and task granularity. For instance, EQ-Bench (referring to its creative writing subset) encompasses templated

Table 1: Comparison of existing writing benchmarks.

| Benchmark | Num | Domains | | Requirement | | | Input Token | | Free Query-Form | Diverse Material-Source |
|---|---|---|---|---|---|---|---|---|---|---|
| | | Primary | Secondary | Style | Format | Length | Avg | Max | | |
| EQ-Bench | 241 | 1 | / | ✗ | ✗ | ✗ | 130 | 213 | ✗ | / |
| LongBench-Write | 120 | 7 | / | ✗ | ✗ | ✓ | 87 | 684 | ✓ | / |
| HelloBench | 647 | 5 | 38 | ✗ | ✗ | ✓ | 1,210 | 7,766 | ✗ | ✗ |
| **WritingBench** | 1,000 | 6 | 100 | ✓ | ✓ | ✓ | 1,699 | 19,361 | ✓ | ✓ |

queries for story-related tasks [21], while LongBench-Write incorporates length constraints in 120 queries [3]; however, they both lack hierarchical domain taxonomies and multi-dimensional requirement specifications (e.g., style and format). Furthermore, most benchmarks rely on fixed instruction templates, short contexts, or materials predominantly from a single source [23, 10, 17], rendering them insufficient for addressing the complexity of real-world needs. In contrast, our proposed benchmark fills these gaps by introducing 1,000 free-form queries distributed across 6 primary domains and 100 subdomains, with potential controls over style, format, and length, paired with inputs ranging from tens to thousands of words.

## 2.2 Evaluation Methods

Using LLMs as evaluators has become a prevalent approach for evaluating the quality of generated responses. Typically, researchers pre-define a fixed set of evaluation dimensions applicable across all test instances [7, 17]. For example, SuperCLUE [30] employs three dimensions, whereas LongBench-Write [3] adopts six dimensions. HelloBench [23] introduces task-specific dimensions, but the dimensions remain consistent across all queries of a given task. Although the LLM-as-a-judge approach enhances scalability, static evaluation dimensions often fail to accommodate the diversity of writing styles and specifications, especially when dealing with inputs with enriched materials. To address this limitation, recent work [15] trains a model to dynamically generate evaluation dimensions for individual queries. However, the dimensions remains confined to a small predefined set. In contrast, our query-dependent evaluation framework leverages LLMs to generate diverse and instance-specific criteria while fine-tuning a dedicated critic model to perform the evaluation.

## 2.3 Writing-Enhanced Models

Although existing LLMs demonstrate exceptional writing capabilities, researchers strive to further enhance their overall writing proficiency. Recent models, such as Weaver [28] and Suri [22], have exhibited notable domain-specific strengths. For instance, Weaver benefits from over 200B parameter pretraining, supporting four distinct writing domains, while LongWriter [3] specializes in length constraints. However, these models experience substantial performance degradation in cross-domain scenarios and multi-constraint tasks. In this work, leveraging the effectiveness of our evaluation framework, we introduce writing-enhanced models trained on high-quality data, achieving high performance across various tasks.

## 3 WritingBench

In this section, we will mainly introduce the four-stage human-AI collaborative construction process of WritingBench, and the query-dependent evaluation framework with its accompanied critic model for criteria-aware evaluation. Additionally, we train a writing-enhanced model to demonstrate the effectiveness of the framework's data curation capability.

## 3.1 Benchmark Construction

WritingBench is constructed following a delicate pipeline (see Figure 2) that combines model-generated query refinement and human annotation, ensuring diversity and real-world relevance. The construction process is illustrated below.

### 3.1.1 Model-Augmented Query Generation

This phase leverages LLMs to generate an initial set of writing queries, which are then enriched and diversified through systematic guidance, with material suggestions provided as needed.

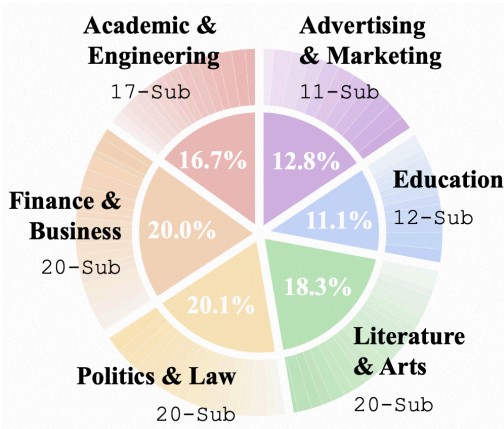

Figure 3: Domain categories in WritingBench. Inner represents the 6 primary domains and outer depicts 100 secondary subdomains (Sub = sub-domains per category).

Table 2: Data statistics for WritingBench.

| Category | Num | Avg Token | Max Token |
|---|---|---|---|
| *Domain* | | | |
| Academic & Engineering | 167 | 1,944 | 15,534 |
| Finance & Business | 210 | 1,867 | 19,361 |
| Politics & Law | 201 | 2,363 | 18,317 |
| Literature & Arts | 183 | 1,266 | 7,675 |
| Education | 111 | 1,345 | 10,737 |
| Advertising & Marketing | 128 | 984 | 6,504 |
| *Requirement* | | | |
| Style | 442 | 1,728 | 19,361 |
| Format | 498 | 1,715 | 15,534 |
| Length | 222 | 1,362 | 14,097 |
| *Length* | | | |
| 0 - 1K | 550 | 470 | 994 |
| 1K - 3K | 292 | 1,832 | 2,991 |
| 3K - 5K | 87 | 3,828 | 4,966 |
| Over 5K | 71 | 8,053 | 19,361 |

**Phase 1 - Initial Query Generation:** We begin by constructing a two-tiered domain pool grounded in real-world writing scenarios, consisting of 6 primary domains and 100 subdomains (see Figure 3 and Appendix B for detailed domain statistics and descriptions; the construction process of domain system is described in Appendix A.1). These domains are designed to capture both traditional and emerging user needs for AI-assisted writing, categorized by topic and functionality into 6 primary domains: Academic & Engineering, Finance & Business, Politics & Law, Literature & Art, Education, and Advertising & Marketing. Leveraging the primary domain and subdomain tags, we prompt two different LLMs (GPT-4o and Claude-3.5-Sonnet [1])to produce an extensive pool of initial writing query drafts that simulate realistic user requests to maximize diversity (see Appendix C.2).

**Phase 2 - Query Diversification:** To enhance the diversity and practical applicability of queries while addressing real-world needs, we design a set of query diversification strategies (see Appendix C.3), inspired by [29]. These strategies are divided into three core requirements and three auxiliary dimensions. The core requirements focus on fundamental aspects of writing: (1) Stylistic adjustments (e.g., "Use a friendly and simple tone that kids can easily understand"), (2) Format specifications (e.g., "Follow the IEEE conference template"), and (3) Length constraints (e.g., "Generate a 500-word executive summary"), which will be evaluated through specialized assessments. The auxiliary dimensions address additional considerations: (4) Personalization (e.g., "From the viewpoint of an educator with two decades of experience"), (5) Content specificity (e.g., "Detail the 2023 Q3 financial metrics"), and (6) Expression (e.g., "Modify the query expression to be shorter"). While refining the queries, the LLM simultaneously provides necessary material suggestions (e.g., requesting financial reports as input for market analysis queries) (see Appendix C.4). This structured approach ensures that queries are both diverse and practical.

### 3.1.2 Human-in-the-Loop Refinement

The above outputs serve solely to provide a large, linguistically diverse pool of drafts. Human experts then verify these queries and supplement the required materials, ensuring alignment with real-world applications and avoiding harmful content.

**Phase 1 - Material Collection:** At this stage, we engage 30 trained annotators, compensated at $18 per hour and possessing specialized expertise proven through domain-knowledge tests. Their role is to collect open-source reference materials needed for the writing tasks (e.g., public financial statements or legal templates). Annotators may either adopt the sources suggested by the LLM-generated material requirements or, when more appropriate, independently identify alternative documents. Each query can be paired with multiple material sources, whereas in some cases no external material is

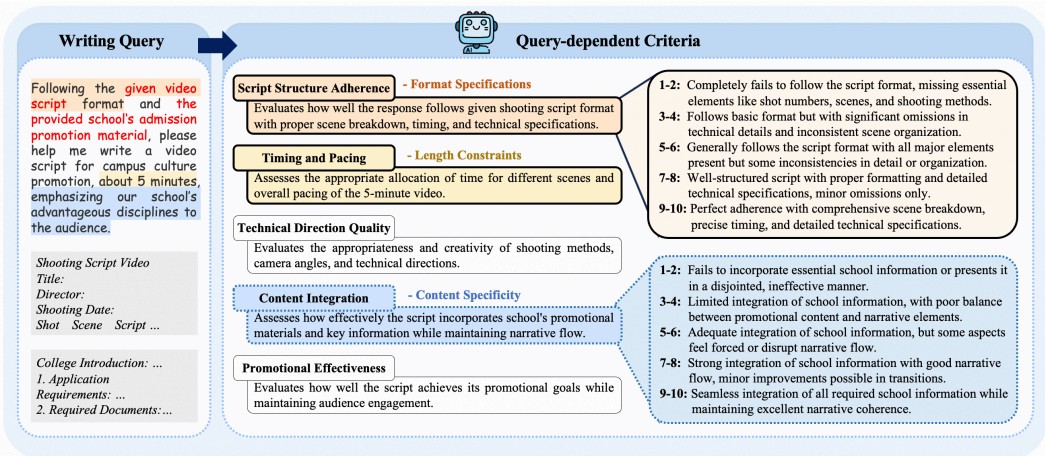

Figure 4: Example of dynamically generating criteria for a writing query in WritingBench. Different background colors represent various types of requirements.

required. To minimize errors caused by parsing documents in varied formats and to eliminate noise from irrelevant content such as web advertisements, the annotators carefully extract and verify only the most pertinent text segments.

**Phase 2 - Expert Optimization:** Next, five experts with LLM experience or relevant industry expertise conduct data screening. The experts first exclude low-quality data that are unrealistic, overly generic, or potentially harmful. For the remaining entries, a rigorous two-stage review is employed: (1) query adaptation: ambiguous expressions are revised to better align with the provided materials and practical scenarios (e.g., adjusting a legal opinion query to reference specific clauses from the supplied statutes), (2) material pruning: redundant or irrelevant information is eliminated from the accompanying materials, ensuring that the context for writing tasks remains focused, relevant, and devoid of harmful content. Furthermore, experts are encouraged to create original, high-quality queries that are cross-validated by their peers, to further enhance diversity and realism.

Finally, we construct WritingBench, a benchmark comprising 1,000 queries categorized using a two-tiered taxonomy, including 445 queries in Chinese and 555 queries in English. In comparison to existing writing benchmarks summarized in Table 1, WritingBench exhibits notable advantages in terms of the number of instances, domain diversity, requirement coverage, and variability in input lengths. The detailed statistical distribution of WritingBench is shown in Table 2.

## 3.2 Query-Dependent Evaluation Framework

Traditional LLM-as-a-judge evaluations typically rely on fixed evaluation criteria derived from general writing assessment conventions [25, 3, 23]. However, such static criteria exhibit three critical limitations: (1) Domain exhaustiveness: fixed criteria are inadequate in adapting to specialized domains, failing to address unique characteristics found in areas like technical documentation or creative writing; (2) Requirement specificity: such criteria lack the flexibility to encompass specific requirements related to style, format, or length control; and (3) Material dependency: they are insufficient to verify whether responses appropriately utilize the provided reference materials, such as incorporating data points or maintaining narrative continuity. To address these challenges, we propose a query-dependent evaluation framework that enables dynamic adaptation to diverse writing scenarios. This approach comprises two phases:

**Phase 1: Dynamic Criteria Generation:** As illustrated in Figure 4, given a query $q$ in the WritingBench, the LLM is prompted to automatically generate a set of five evaluation criteria, $C_q = \{c_1, \ldots, c_5\}$ (opting for five criteria reflects a common approach observed in many contemporary evaluation contexts [25, 16, 17]), using a carefully designed instruction to ensure structural guidance during criteria specification (see Appendix C.5). We utilize Claude-3.7 for generation, as it demonstrates superior diversity and comprehensiveness in criteria generation compared to models

Table 3: WritingBench performance of LLMs across 6 domains and 3 core requirements evaluated with our critic model (scale: 1-10). The standard deviation is computed over 3 samples. Domains include: (D1) Academic & Engineering, (D2) Finance & Business, (D3) Politics & Law, (D4) Literature & Art, (D5) Education, and (D6) Advertising & Marketing. The writing requirements assessed are: (R1) Style, (R2) Format, and (R3) Length. Here, "C" indicates category-specific scores. The latest results are available on the online leaderboard.

| Models | Overall | Languages | | Domains | | | | | | Requirements | | | | | |
|---|---|---|---|---|---|---|---|---|---|---|---|---|---|---|---|
| | | ZH | EN | D1 | D2 | D3 | D4 | D5 | D6 | R1 | C | R2 | C | R3 | C |
| *Proprietary LLMs* | | | | | | | | | | | | | | | |
| Claude-3.7-thinking | $7.91_{\pm 0.111}$ | 7.9 | 7.9 | 7.9 | 7.8 | 7.8 | 8.0 | 8.0 | 8.1 | 7.9 | 8.7 | 8.0 | 8.4 | 8.0 | 8.1 |
| Claude-3.7 | $7.85_{\pm 0.110}$ | 7.9 | 7.8 | 7.8 | 7.8 | 7.7 | 7.9 | 8.0 | 8.1 | 7.9 | 8.6 | 7.9 | 8.3 | 8.0 | 8.1 |
| Qwen-Max | $7.16_{\pm 0.041}$ | 7.2 | 7.1 | 7.1 | 6.9 | 7.0 | 7.3 | 7.4 | 7.5 | 7.2 | 8.3 | 7.3 | 7.8 | 7.2 | 7.5 |
| o1-Preview | $6.89_{\pm 0.039}$ | 6.8 | 7.0 | 6.9 | 6.8 | 6.7 | 7.0 | 7.1 | 7.2 | 6.9 | 8.0 | 7.0 | 7.5 | 7.1 | 7.3 |
| GPT-4o | $6.81_{\pm 0.028}$ | 6.9 | 6.7 | 6.8 | 6.6 | 6.7 | 6.8 | 7.0 | 7.1 | 6.9 | 8.0 | 7.0 | 7.5 | 6.8 | 6.8 |
| Gemini-1.5-Pro | $6.21_{\pm 0.018}$ | 6.2 | 6.2 | 6.2 | 5.8 | 6.0 | 6.4 | 6.6 | 6.7 | 6.2 | 7.2 | 6.4 | 7.1 | 6.4 | 6.0 |
| *Open-source LLMs* | | | | | | | | | | | | | | | |
| Deepseek-R1 | $7.70_{\pm 0.053}$ | 8.0 | 7.5 | 7.6 | 7.4 | 7.6 | 7.8 | 7.8 | 8.1 | 7.7 | 8.4 | 7.9 | 8.3 | 7.7 | 7.5 |
| Deepseek-V3 | $6.35_{\pm 0.022}$ | 6.3 | 6.4 | 6.4 | 6.1 | 6.2 | 6.3 | 6.6 | 6.8 | 6.4 | 7.6 | 6.5 | 7.1 | 6.5 | 6.4 |
| Mistral-Large-Instruct | $6.00_{\pm 0.076}$ | 5.9 | 6.1 | 6.2 | 5.9 | 5.9 | 5.7 | 6.4 | 6.4 | 6.1 | 7.3 | 6.1 | 6.5 | 6.0 | 6.0 |
| Qwen-2.5-72B-Instruct | $6.40_{\pm 0.061}$ | 6.4 | 6.4 | 6.6 | 6.2 | 6.4 | 6.2 | 6.7 | 6.6 | 6.5 | 7.7 | 6.5 | 6.9 | 6.5 | 6.5 |
| Qwen-2.5-7B-Instruct | $5.64_{\pm 0.083}$ | 5.5 | 5.8 | 5.9 | 5.6 | 5.6 | 5.1 | 6.1 | 5.9 | 5.7 | 7.0 | 5.7 | 6.1 | 5.6 | 5.6 |
| Llama-3.3-70B-Instruct | $5.05_{\pm 0.011}$ | 4.5 | 5.5 | 5.1 | 4.9 | 4.8 | 4.8 | 5.3 | 5.6 | 5.0 | 5.0 | 5.1 | 5.9 | 5.1 | 5.0 |
| Llama-3.1-8B-Instruct | $4.42_{\pm 0.004}$ | 3.7 | 5.0 | 4.1 | 4.4 | 4.0 | 4.1 | 4.7 | 5.0 | 4.4 | 4.4 | 4.5 | 5.3 | 4.4 | 4.3 |
| *Capability-enhanced LLMs* | | | | | | | | | | | | | | | |
| Suri | $3.20_{\pm 0.042}$ | 2.5 | 3.8 | 3.6 | 3.5 | 3.0 | 2.5 | 3.2 | 3.6 | 3.2 | 3.7 | 3.1 | 3.2 | 3.0 | 3.0 |
| LongWriter | $6.27_{\pm 0.081}$ | 6.2 | 6.4 | 6.4 | 6.4 | 6.3 | 6.0 | 6.5 | 6.0 | 6.3 | 7.4 | 6.3 | 6.7 | 6.3 | 6.8 |
| Qwen-2.5-7B-filtered | $7.44_{\pm 0.058}$ | 7.7 | 7.2 | 7.4 | 7.2 | 7.5 | 7.3 | 7.7 | 7.7 | 7.5 | 8.4 | 7.6 | 8.1 | 7.4 | 7.2 |
| Llama-3.1-8B-filtered | $7.39_{\pm 0.045}$ | 7.5 | 7.3 | 7.4 | 7.2 | 7.3 | 7.3 | 7.5 | 7.8 | 7.4 | 8.3 | 7.5 | 8.0 | 7.4 | 7.1 |

such as GPT-4o(see Appendix A.5.2 for details). Each criterion comprises three components: a concise name summarizing the criterion, an extended description elaborating on the evaluation focus, and detailed scoring rubrics, which provide fine-grained quality levels for the respective evaluation dimensions. The generated criteria are further reviewed by human annotators to confirm their reasonableness and ensure no harmful content.

We identify three commonly encountered writing requirements: style, format, and length from real-world writing scenarios (frequency analysis detailed in Appendix A.1). For each requirement, we create two evaluation subsets. Two annotators review each query to classify each criteria in $C_q$ into style, format, length, or none. Based on their annotations, we perform final verification and define the subsets as follows: (1) The first requirement-involved subset includes all queries involving a specific requirement and their entire set of criteria. For example, if any criterion in $C_q$ relates to style, the query and its entire criteria set $C_q$ belong to this subset. Scores for all five criteria are calculated, corresponding to the R1/R2/R3 columns in Table 3. (2) The second category-specific subset includes only the criteria related to the specific requirement. For instance, if $c_1$ and $c_4$ in $C_q$ are format-related, then the query and the subset $c_1, c_4$ belong to this subset. Only the scores for $c_1, c_4$ are calculated, corresponding to the C columns. The first subset serves as an overall evaluation of writing quality for a specific requirement, while the second subset provides more targeted insights into performance on that particular capability.

**Phase 2 - Rubric-based Scoring:** For each criterion $c_i \in C_q$, the evaluator independently assigns a score on a 10-point scale to a response $r$, along with a justification. The final overall score is computed by averaging scores across all criteria. Scoring prompt is provided in Appendix C.6.

To alleviate the computational overhead with LLM-based evaluation, we develop a dedicated critic model, $\mathcal{M}$, designed to implement our rubric-based scoring framework. Specifically, this model performs the mapping $\mathcal{M}_c : (q, r, c_i) \mapsto [1, 10] \times \mathcal{J}$, where the output consists of a numerical score and corresponding justification text, $\mathcal{J}$, both in accordance with the predefined evaluation rubric. The critic model is fine-tuned on a dataset of 155K instances scored by Claude-3.7, which demonstrates higher human alignment consistency (as discussed in Section 4.3). When building the training data of scoring prompts, the queries and criteria are drawn from WritingBench. The response portion is generated using approximately 40 different models, including those evaluated in

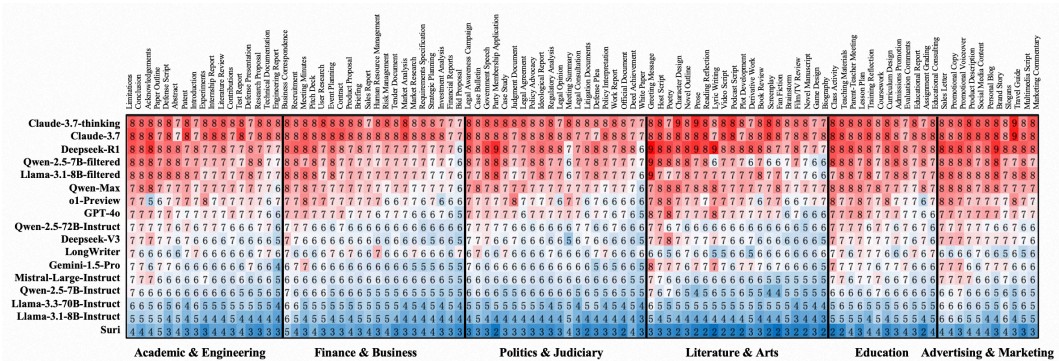

Figure 5: Score heatmap of models across 100 subdomains in WritingBench. The figure shows the average score per subdomain for each model. Red indicates higher score and blue indicates lower.

Section 4.2, as well as additional models of varying types and sizes, such as Claude-3.5-Haiku [1] and Llama-3.2-1B-Instruct [18]. Finally, after removing instances where scoring failed, approximately 155K samples are constructed for training. The experiments described in Section 4.3 confirm the consistency of the critic model.

### 3.3 Evaluation-Guided Data Curation for Writing Enhancement

The query-dependent evaluation framework enables systematic data curation across diverse writing tasks through two-phase filtering. To validate its effectiveness in data curation, we conduct SFT experiments using constructed datasets. We utilize the secondary domain taxonomy of WritingBench and follow the first two steps outlined in Section 3.1.1 (removing the material generation suggestion part from the prompt in Appendix C.4, requiring only the refined query from the model) to let LLMs generate writing queries. To enrich query diversity, we employ two models, GPT-4o and Claude-3.5-Sonnet, to each generate 60 queries in both Chinese and English for every subdomain, totaling 24K queries. Responses are uniformly generated using an advanced model, Deepseek-R1.

Subsequently, we apply the query-dependent evaluation metric with our critic model described in Section 3.2 to select top-50% samples per subdomain, resulting in 12K high-quality data samples. Fine-tuning experiments are conducted using the Llama-3.1-8B-Instruct and Qwen-2.5-7B-Instruct models. Both models (Qwen-2.5-7B-filtered and Llama-3.1-8B-filtered) demonstrated significant performance improvements over their base versions and even outperformed larger models such as Llama-3.3-70B-Instruct and Qwen-2.5-72B-Instruct in our experiments on two benchmarks, confirming the framework's effectiveness in identifying high-quality writing samples.

## 4 Experiment

### 4.1 Experiment Settings

This section outlines the settings employed in our experiments using the WritingBench framework. The evaluation protocol and the model training configurations are as follows:

**Evaluation Protocol:** Uniform generation settings are applied to model responses on WritingBench, with maximum sequence lengths capped at 16,000 tokens (or platform maximum if lower). We configure the generation process with a temperature of 0.7, a top-k value of 20, and top-p of 0.8, ensuring a balance of creativity and coherence. For both the critic model and LLM scoring, we use consistent configurations. The temperature is set to 1.0, in line with the Arena-Hard-Auto framework [14]. We apply a top-p value of 0.95 without any top-k filtering, and the maximum length is set at 2,048 tokens to ensure efficient and reliable output generation.

**Model Training:** The critic model is fine-tuned on the Qwen-2.5-7B-Instruct base model, using the AdamW optimizer with a 7e-6 learning rate. It is trained on 155K SFT data for 3 epochs across 16xA100 GPUs (batch size 64, 4-step accumulation). The writing models, trained on both Qwen-2.5-7B-Instruct and Llama-3.1-8B-Instruct for 5 epochs on 32xA100 GPUs, achieving a total batch size of 128 with 4-step gradient accumulation.

Table 4: Comparison of human consistency scores across different criteria generation methods. Claude corresponds to Claude-3.7. Critic refers to the critic model.

| Evaluation Metric | Judge | Score |
|---|---|---|
| Static Global | GPT-4o | 69% |
| Static Domain-Specific | GPT-4o | 40% |
| Dynamic Query-Dependent | GPT-4o | 79% |
| Static Global | Claude | 67% |
| Static Domain-Specific | Claude | 58% |
| **Dynamic Query-Dependent** | Claude | 87% |
| **Dynamic Query-Dependent** | **Critic** | 84% |

Table 5: Performance of our writing models on two benchmarks. '-filtered' indicates models trained with filtered data, while '-all' uses the full dataset. Eval2 refers to LongBench-Write [3].

| Models | WritingBench | Eval2 |
|---|---|---|
| Deepseek-R1 | 7.70 | 4.79 |
| Qwen-2.5-7B-Instruct | 5.64 | 4.39 |
| Llama-3.1-8B-Instruct | 4.42 | 3.12 |
| Qwen-2.5-7B-all | 7.36 | 4.69 |
| Qwen-2.5-7B-filtered | 7.44 | 4.70 |
| Llama-3.1-8B-all | 7.34 | 4.65 |
| Llama-3.1-8B-filtered | 7.39 | 4.65 |

## 4.2 Comparison between LLMs

We evaluate 17 LLMs: GPT-4o[2] [8], o1-Preview [9], Claude-3.7 and its thinking version Claude-3.7-thinking [2], Gemini-1.5-Pro [24], Qwen-Max [26], Deepseek-R1 [4], Deepseek-V3 [5], Mistral-Large-Instruct [20], Qwen-2.5-72B-Instruct and Qwen-2.5-7B-Instruct [31], Llama-3.3-70B-Instruct and Llama-3.1-8B-Instruct [6], Suri [22], LongWriter [3], and our writing model, Qwen-2.5-7B-filtered (fine-tuned on Qwen-2.5-7B-Instruct) and Llama-3.1-8B-filtered (fine-tuned on Llama-3.1-8B-Instruct), on WritingBench. For models capable of reasoning, the reasoning content is excluded from the evaluated response. Each query is assessed by 5 instance-specific criteria using a 10-point scale scoring by our critic model. The overall scores of the models are presented in Table 3, along with their performance across two languages and various domain and requirement subsets. Detailed variations are further revealed through subdomain-specific subcategory heatmap in Figure 5.

**Key Insights from Domain Scores:** The Education (D5) and Advertising & Marketing (D6) domains always yield substantial performance across models. In contrast, the Academic & Engineering (D1) and Finance & Business (D2) domains present greater challenges, as these tasks inherently require more sophisticated information processing and integration capabilities. Our evaluation across 100 subdomains further identifies persistent difficulties in niche areas like writing bid proposals, financial reports, and white papers, where models generally achieve lower scores. These tasks demand a higher level of knowledge, long-text generation capabilities, and adherence to contextual consistency, pinpointing areas for further enhancement.

The Literature & Art (D4) domain exhibits notable performance variance among models. Reasoning-capable architectures such as Claude-3.7-thinking, Deepseek-R1, and o1-Preview outperform their non-reasoning counterparts, indicating the potential of CoT techniques in processing narrative and creative content. To further validate CoT's efficacy in creative writing, we employ the 12k SFT dataset described in Section 3.3 to fine-tune Qwen-32b-Instruct using both CoT-integrated and non-CoT approaches. Evaluations are conducted on the Literature & Art (D4) subset of WritingBench and EQ-Bench [21], a specialized benchmark designed for creative writing tasks. The results reveal that the reasoning model consistently surpass both the baseline Qwen-32b-Instruct and its non-reasoning variants, with detailed results provided in Appendix A.2.

**Key Insights from Requirement Scores:** Most models perform well in the style dimension, followed by format, with length being the weakest. We observe that advanced models often achieve higher specialized scores (C column) for the three common requirements compared to their overall scores (R column). Criteria outside these specialized sets tend to emphasize content-related aspects, such as integration with source materials and writing depth, underscoring the need to improve content quality.

Length requirements remain particularly difficult, especially in section-specific constraints (see Appendix 7 for examples) and extended text generation. Additionally, we evaluate the performance of LLMs across varying input and output lengths (see Appendix A.4 for details). Advanced models generally sustain consistent performance across varying input lengths, leveraging their strong long-context comprehension abilities. However, regarding output length, most models demonstrate inherent limitations, typically capping their responses at around 3,000 tokens. In contrast, Claude-3.7 and its

---

[2]In this paper, we specifically use GPT-4o, version gpt-4o-2025-01-29.

reasoning-enhanced version, as well as Qwen-Max, stand out for their capacity to generate extended outputs effectively. These results emphasize the critical need to improve long-output generation and refine length optimization in writing-related tasks.

The overall analysis of WritingBench experiment highlights: (1) Claude-3.7-thinking consistently leads across both domain and requirement dimensions, followed by its non-reasoning variant, showcasing versatility and strong language capabilities; (2) A significant performance variance is observed within creative content domains, where models incorporating CoT mechanisms surpass those without, demonstrating CoT's potential in LLM writing; (3) Cross-lingual inconsistencies in models like Deepseek-R1 and Llama-3.3-70B-Instruct suggest limitations in multilingual knowledge alignment. Furthermore, detailed analysis on subdomains reveals persistent difficulties in knowledge-intensive tasks. Ablation experiments on output length emphasize that generating long-form outputs remains an obstacle for current models. This analysis not only benchmarks the existing capabilities of these models but also underscores specific areas needing improvement for future development. For the most up-to-date results, please refer to the online leaderboard[3].

### 4.3 Human Consistency

To validate the alignment between automated evaluation and human judgment, we conduct a human evaluation study involving 300 queries (constructed in the same pipeline as described in Section 3.1 but not included in WritingBench). Five professionally trained annotators with linguistic backgrounds perform pairwise comparisons of model responses. For each query, two responses are generated by two randomly selected models (drawn from the same set of 17 models used for evaluation in Section 4.2). The triplet <Query, Response_A, Response_B> is presented on the same page to enable direct comparison. Annotators are instructed to carefully read the query and select their preferred response or declare equivalence (A/B/Tie), with no criteria information provided to ensure unbiased decision-making (detailed instructions and the annotation interface are provided in Appendix A.3).

We then compare these human preferences with the scores assigned by our critic model to the responses — if the critic model rate Response_A higher and the annotators also prefer Response_A, it is counted as alignment. To further evaluate the effectiveness of our dynamic query-dependent criteria, we compare it against two baselines: static globally uniform criteria and static domain-specific customized criteria (designed by domain experts). This comparison was conducted using two LLM-based judges, GPT-4o and Claude-3.7. Both models score the responses using the rubric-based scoring method described in Section 3.2, using scoring prompts provided in Appendix C.6.

As shown in Table 4, our dynamic query-dependent criteria achieve superior human alignment compared to static, both globally uniform or domain-specific customized criteria. Notably, domain-specific criteria underperform despite customization, since our queries involve highly diverse tasks and varied sources. These findings confirm that context-sensitive query-dependent evaluation better captures real-world writing complexity compared to conventional static approaches. Furthermore, the critic model attains 84% agreement, confirming its practical viability.

### 4.4 Ablation of Data Curation for Writing-Enhanced Models

To validate the data curation capabilities of the query-dependent evaluation framework described in Section 3.2, we conduct fine-tuning experiments on two datasets: the initial 24K dataset constructed in Section 3.3 and the 12K subset curated using the query-dependent evaluation framework. We experiment with two models of different architectures, Llama-3.1-8B-Instruct and Qwen-2.5-7B-Instruct, and evaluate their performance on two benchmarks: WritingBench and LongBench-Write, a general-purpose writing benchmark (following the quality evaluation settings outlined in [3]).

As shown in Table 5, both models trained on the filtered 12K dataset demonstrate significant performance improvements over their previous versions. Notably, they outperform models trained on the full 24K dataset and even approach the capabilities of advanced models. These results validate the robustness of our query-dependent evaluation strategy and highlight the effectiveness of our critic model in curating high-quality writing samples. This integrated approach enables smaller models to compete with, and in some cases surpass, larger models across a wide range of writing tasks.

---

[3]For clearer display, online leaderboard scores use the same calculations but are rescaled to a 100-point scale.

# 5 Conclusion

In this paper, we introduce WritingBench, a comprehensive benchmark designed to evaluate LLMs in generative writing across diverse domains. It includes 1,000 queries spanning 6 primary domains and 100 subdomains, providing evaluation dimensions for style, format, and length requirements. Our query-dependent evaluation framework, supported by a critic model, achieves high human alignment. Evaluation efficiency is further demonstrated by compact models trained on curated data, outperforming GPT-4o in writing. By making WritingBench and its resources publicly available, we aim to foster further research and advancements in LLM writing capabilities.

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

# A Experiment Results

## A.1 Validation of Domain Taxonomy Construction and Requirement Dimension

With existing benchmarks focusing on limited tasks (e.g. fiction), the domain taxonomy of our benchmark is rigorously designed to reflect real-world writing scenarios. We leverage the industrial background of team members to refine the domain taxonomy using over 200K anonymized real-user writing query data under strict data security protocols. Our initial two-tier domain system is established, referencing the mature workflows of industry product teams, with an additional "Other" category under each primary domain. This is iteratively refined through multiple rounds: in each iteration, we use the prompt outlined in Appendix C.1 and perform classification using GPT-4o to tag 2K randomly selected writing queries data according to the current subdomain labels, with each query potentially receiving multiple tags. Subsequently, human annotation validate the tags and adjust the domain settings for the next iteration, aiming to reduce overlaps and minimize the "Other" category. The final tags achieve stable after three iterations, ensuring broad domain coverage.

To evaluate the classification accuracy of GPT-4o, we validate the effectiveness of the model's automatic annotations on a writing task tag system provided by the production team. This tag system consists of 8 primary domains and 44 subdomains, with a dataset of 3,000 entries that have been manually annotated and verified under strict data security protocols. The recall rate is calculated such that if a manually assigned tag is found within the list of tags generated by the model, it is considered successfully retrieved. The overall recall rate for primary domains is approximately 96%, while the recall rate for subdomains is about 93%.

By analyzing the queries provided by the production team, we identify three prevalent requirement dimensions: style, format, and length. To validate these categories, we perform model-based annotations across several iterations, using the same prompt outlined in Appendix C.1. After multiple rounds of analysis, the average frequencies are determined as follows: style-related requirements account for approximately 29.4%, format-related requirements comprise about 22.48%, and length-related requirements make up around 20.8%. The remaining requirements largely correspond to criteria closely associated with writing materials or other demands that are challenging to generalize into abstract dimensions.

## A.2 Ablation of CoT in Creative content

To validate the impact of CoT reasoning on creative content generation, we conduct SFT experiments on filtered subset described in Section 3.3 on Qwen-2.5-32B-Instruct. Two variants are developed: 1) A base model trained with CoT-formatted instructions, adhering to DeepSeek-R1's original output format, and 2) an ablated version (-w/o CoT) trained using only the response content. These models are evaluated on the Literature & Art (D4) subset of WritingBench and EQBench, a specialized benchmark for creative writing evaluation, to thoroughly examine their capabilities in generating creative content.

Table 6: Ablation of CoT in creative content on two benchmarks.

| Models | WritingBench-D4 | EQBench |
|---|---|---|
| Deepseek-R1 | 7.70 | 84.99 |
| Qwen-2.5-32B-Instruct | 5.59 | 48.17 |
| Qwen-2.5-32B-CoT | 7.58 | 82.48 |
| - w/o CoT | 7.54 | 79.43 |

As shown in Table 6, the CoT-enhanced model outperform the non-reasoning ones on both WritingBench-D4 and EQBench, showing CoT's effectiveness in generating creative content.

## A.3 Human Consistency

Human evaluations are conducted via DingTalk Docs (a cloud-based collaborative platform[4]), where annotators access standardized spreadsheets. The spreadsheets contain triplets of <Query, Response_A, Response_B> with queries highlighted in yellow and instructions are provided in blue header sections (refer to the interface screenshot in Figure 6). To mitigate positional bias, response ordering is randomized per instance, and annotators could zoom into cells for full-text review. The interface enable keyword searches across responses and queries for systematic comparison. The test set excludes inputs exceeding 5,000 tokens to reduce cognitive load, with explicit instructions emphasizing content quality over text length or surface formatting. Five linguists achieved substantial agreement ($\kappa = 0.69$), demonstrating rigorous annotation reliability.

Figure 6: Annotation Interface.

## A.4 Ablation of Length

We assess the performance of LLMs across varying input and output lengths (see Figure 7), with statistical validity ensured by excluding intervals containing fewer than 5 samples. Experiments on input length reveal that most SOTA models generally maintain consistent performance regardless of input length variations, attributable to their advanced long-context comprehension capabilities. However, analysis on output length shows that some models exhibit inherent limitations in response generation length, typically producing outputs constrained to approximately 3,000 tokens. Small models, such as Suri, Qwen-2.5-7B-Instruct, Llama-3.1-8B-Instruct, suffer more performance degradation characterized by repetitive outputs. Notably, Claude-3.7 and its reasoning model, Qwen-Max and LongWriter effectively support extended response lengths. These findings highlight the importance of improving long-output generation capabilities and optimizing length handling in writing tasks.

---

[4]https://www.dingtalk.com/en

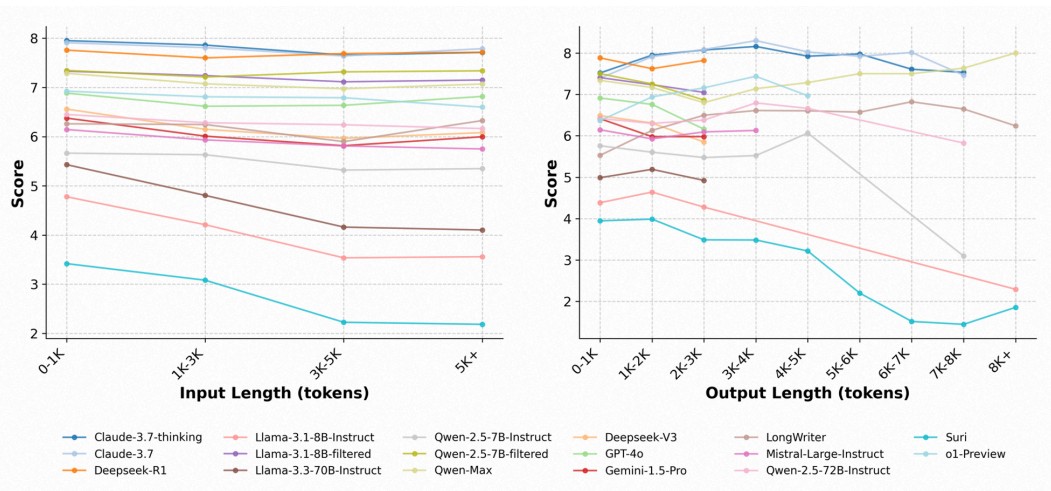

Figure 7: Scores of different models across various input and output lengths on the WritingBench.

## A.5 Case Study

### A.5.1 Bad Case of Length Constraint

Current models often face challenges when handling length-related requirements. Figure 7 illustrates bad cases where the LLM struggle with adhering to length constraints.

Table 7: Bad case of length requirement. Query from WritingBench with index 539, response generated by Gemini-1.5-Pro.

| Query |
| --- |
| Create advertising slogans for a pet grooming salon, including: |
| - Main slogan (10-15 words) |
| - Sub-slogan (15-20 words) |
| - Service tagline (8-12 words) |
| - Brand promise (10-15 words) |
| The slogans should reflect professional pet grooming and care services while emphasizing thoughtful care for pet owners. The slogans should be catchy, memorable, with warm and friendly wording, and a light, cheerful tone. |
| **Response** |
| **Main Slogan:** Pamper Your Pet, Delight Your Heart. |
| **Sub-Slogan:** Expert grooming and loving care, making your furry friend feel fabulous, inside and out. |
| **Service Tagline:** Gentle paws, happy pets, stress-free smiles. |
| **Brand Promise:** Quality grooming, compassionate care, every time. |

### A.5.2 Criteria Generation Comparison

We compare the ability of various models to generate criteria and ultimately selected Claude-3.7, which demonstrates advantages in diversity, comprehensiveness, and rationality of the criteria. Figure 8 presents a comparison between Claude-3.7 and GPT-4o based on the same example query. The criteria descriptions generated by Claude-3.7 show a higher degree of integration with the specific requirements of the query. In contrast, we observe in more cases that GPT-4o tends to uses similar criteria and has a lower level of integration with the query, such as not adequately considering information about the material.

**Query:**
Based on the following article, please help me write an introduction with the following requirements:
Introduction:
Research Purpose:
Research Content:
Research Subjects:  fashion design studio and students from university
Research Methods: In-person interviews and questionnaires.
Research Timeframe: April 20 to May 5."

… (the remaining chapters of an completed report.)

**Criteria generated by Claude-3.7**
```
[
    {
        "name": "Structural Completeness",
        "criteria_description": "Evaluates whether the preface includes all required components (research purpose, content, subjects, methods, and timeframe) in a well-organized manner",
    },
    {
        "name": "Research Context Clarity",
        "criteria_description": "Assesses how clearly the response articulates the research context regarding the fashion design studio and student subjects",
    },
    {
        "name": "Methodological Precision",
        "criteria_description": "Evaluates the clarity and specificity of research methods description (interviews and questionnaires)",
    },
    {
        "name": "Temporal Framework",
        "criteria_description": "Assesses how well the timeline is integrated and explained within the research context",
    },
    {
        "name": "Professional Tone",
        "criteria_description": "Evaluates the overall professional and academic quality of the preface's language and presentation",
    }
]
```

**Criteria generated by GPT-4o**
```
[
    {
        "name": "Relevance to Query",
        "criteria_description": "Evaluates how closely the response aligns with the query requirements, ensuring it directly",
    },
    {
        "name": "Coherence and Organization",
        "criteria_description": "Assesses the logical flow, readability, and structure of the response, ensuring it presents information in an organized manner.",
    },
    {
        "name": "Depth of Content",
        "criteria_description": "Measures the thoroughness of the response, analyzing whether it provides detailed and substantial information as required by the query.",
    },
    {
        "name": "Specificity and Precision",
        "criteria_description": "Analyzes the degree of specificity and precision in the response, ensuring it avoids vague or overly general information.",
    },
    {
        "name": "Adherence to Query Context",
        "criteria_description": "Evaluates how well the response adheres to the context of the query, including the tone, format, and inclusion of required subsections.",
    }
]
```

Figure 8: Compasion of criteria generated by Claude-3.7 and GPT-4o.

# B  Benchmark Statistics

## B.1  Overview of Six Primary Domains

1. **Academic & Engineering:** This domain encompasses the structured and formalized nature of academic writing workflows, focusing on clarity, precision, and adherence to rigorous standards. includes the creation of paper outlines, abstracts, literature reviews, experiment reports, and technical documents such as patents and test reports. The writing prioritizes logical argumentation, thorough analysis, and the integration of empirical evidence.

2. **Finance & Business:** Writing in this domain is analytical and strategic, aimed at informing decision-making and promoting corporate objectives. It includes a wide range of documentation such as contracts, market analyses, investment reports, strategic plans, and operational materials like product specifications and sales reports. The emphasis is on clarity and conciseness, with a focus on financial acumen and strategic insights.

3. **Politics & Law:** This domain demands an authoritative and formal tone, as it involves the composition of government documents, legal writings, and political communications. These materials require a careful balance between clarity and formality, often employing complex and structured language. The aim is to clearly convey policy positions, legal arguments, or political messages while strictly adhering to legal and procedural standards.

4. **Literature & Art:** This domain covers the creative and expressive realms of writing, including novels, poetry, scripts, artistic designs, and critiques of books and movies. Writers explore thematic and emotional depths, crafting works that connect with audiences on a human level. The language is rich and evocative, allowing for a personal exploration of ideas that engage and move the reader.

5. **Education:** This domain involves pedagogical materials and educational communication, including lesson plans, course designs, feedback, assignments, and institutional communications like admissions promotions and parent-teacher meeting scripts. The writing prioritizes clarity, accessibility, and instructional effectiveness, using an approachable tone to facilitate learning and engagement.

6. **Advertising & Marketing:** Writing in this domain is vibrant and persuasive, designed to captivate and influence target audiences across various digital platforms. It includes social media scripts, advertising copy, brand narratives, and multimedia campaign materials. The writing is dynamic and strategic, with a creative twist, necessitating a deep understanding of audience psychology and trend dynamics to ensure that the content is appealing and strategically effective.

## B.2  Overview of 100 Secondary Subdomains

See Table 8, Table 9 and Table 10.

Table 8: Subdomains in Academic & Engineering and Finance & Business.

| SubDomain | Description |
|---|---|
| *Academic & Engineering* | |
| Paper Outline | Hierarchical organization of research components and logical flow |
| Acknowledgments | Formal recognition of institutional and individual support |
| Limitations | Systematic identification of methodological constraints and scope boundaries |
| Defense Presentation | Presentation supporting materials, such as slides |
| Research Proposal | Investigation blueprint with validation road map |
| Technical Documentation | Implementation specifications and system interface protocols |
| Experiments | Parameterized validation framework with controlled variable analysis |
| Introduction | Contextual foundation establishing research gaps and significance |
| Conclusion | synthesize the main findings of the research or project |
| Test Report | Evaluations of testing activities and performance |
| Contributions | Novel aspects differentiating the work from prior research |
| Internship Report | Chronological documentation of a practical work placement |
| Literature Review | Critical gap analysis through scholarly works taxonomy |
| Defense Script | Oral presentations and responses for research defense. |
| Abstract | Summary of research objectives, methods, results, and significance |
| Engineering Report | Technical analysis on tasks, methodologies, and outcomes |
| Patent | Legal-technical specification of novel implementable claims |
| *Finance & Business* | |
| Meeting Minutes | Concise documentation of key discussion points, decisions, and action items |
| User Research | Insight collection on user needs and behaviors to inform product or service design |
| Business Correspondence | Formal communication with internal or external stakeholders for business purposes |
| Human Resource Management | Strategies and processes for managing workforce effectively |
| Recruitment | Strategies for attracting, selecting, and onboarding suitable candidates |
| Briefing | Summarized information provided to stakeholders ahead of a task or meeting |
| Event Planning | Coordinated organization of logistics and activities for event execution |
| Market Research | Systematic collection and analysis about market and consumer |
| Market Analysis | Evaluation of market trends, size, competitors, and dynamics |
| Risk Management | Identification, assessment, and prioritization of risks with mitigation strategies |
| Sales Report | Summary of sales activities, performance, and revenue over a specific period |
| Pitch Deck | Visual presentation for communicating business ideas or proposals to investors |
| Contract | Legally binding agreement detailing terms and conditions for business transactions |
| Tender Document | Formal proposal request containing project specifications and bidding instructions |
| Investment Analysis | Evaluation of financial investments to determine potential returns and risks |
| Product Proposal | Detailed plan outlining the development, features, and potential of new products |
| Strategic Planning | Business goal setting with actionable strategies for desired outcomes |
| Financial Reports | Comprehensive statements reflecting the financial performance and status |
| Requirements Specification | Documentation detailing functional and non-functional requirements for a project |
| Bid Proposal | Formal offer to supply goods or services at a set price, meeting client needs |

Table 9: Subdomains in Politics & Law and Literature & Art.

| Subdomain | Description |
| --- | --- |
| *Politics & Law* | |
| Legal Opinion | Authoritative assessment and guidance on legal matters or questions |
| Government Speech | Formal address by government officials outlining policies or positions |
| Judgment Document | Official written decision or order issued by a court |
| Legal Agreement | Binding contract setting out terms and obligations between parties |
| Case Study | In-depth analysis of a legal case for educational or professional purposes |
| Case Bulletin | Summary and update on ongoing or concluded legal cases |
| Legal Consultation | Professional advice provided on legal rights, responsibilities, or strategies |
| Regulatory Analysis | Examination of rules and regulations affecting compliance and enforcement |
| Meeting Summary | Brief overview of discussions, decisions, and outcomes from a meeting |
| Ideological Report | Analysis or commentary on political or ideological trends and perspectives |
| Policy Interpretation | Explanation or clarification for public or organizational guidance |
| Official Document | Formal written record issued by government entities or officials |
| Legal Awareness Campaign | Initiative to educate the public on legal rights and responsibilities |
| Defense Plea | Formal written argument submitted by the defense in a legal proceeding |
| Party Membership Application | Form and process for joining a political party |
| Policy Advocacy | Efforts to influence or promote specific policy changes or implementations |
| Work Report | Report of activities, achievements, and challenges within a specific period |
| Deed Achievement | Record highlighting significant accomplishments and contributions |
| Litigation Documents | Legal filings and paperwork submitted in the course of a lawsuit |
| White Paper | Authoritative report providing information or proposals on an issue |
| *Literature & Art* | |
| Character Design | Creation and development of detailed characters for stories or visual media |
| Greeting Message | Friendly or formal introductory statement used for various occasions |
| Host Script | Guided narration and dialogue for a presenter during an event or show |
| Novel Outline | Structured plan for the plot, characters, and settings of a novel |
| Podcast Script | Written content outlining the dialogue and segments for podcast episodes |
| Derivative Work | Creative work based on or inspired by an existing piece |
| Reading Reflection | Personal thoughts and analysis on a piece of literature |
| Video Script | Script detailing dialogue and action for video content creation |
| Book Review | Critical evaluation and summary of a book's content and impact |
| Game Design | Creation of mechanics, stories, and interfaces for games |
| Lyric Writing | Crafting of words for songs with rhyme and meter considerations |
| Brainstorm | Rough ideas and notes generated during a creative thinking session |
| Plot Development | Process of mapping out the storyline and narrative structure |
| Prose | Written or spoken language in its ordinary form, without metrical structure |
| Screenplay | Scripted blueprint for film or television with dialogue and directions |
| Novel Manuscript | Complete text of a novel prepared for publication |
| Biography | Detailed account of a person's life experiences and achievements |
| Film/TV Review | Analytical critique of a film or television show's content and effectiveness |
| Poetry | Artistic composition using rhythmic and metaphorical language |
| Fan Fiction | Amateur stories by enthusiasts featuring characters from existing media |

Table 10: Subdomains in Education and Advertising & Marketing.

| SubDomain | Description |
|---|---|
| *Education* | |
| Training Reflection | Personal assessment of training experiences and learned insights |
| Class Activity | Planned exercises or tasks designed to engage students in learning |
| Parent-Teacher Meeting | Formal discussion between educators and parents about student progress |
| Lesson Plan | Structured outline of educational objectives and teaching methods |
| Teaching Materials | Resources used to aid in presenting information to students |
| Assignment Grading | Evaluation and scoring of student work based on specific criteria |
| Curriculum Design | Development of educational content, structure, and delivery methods |
| Educational Report | Analysis or summary of educational outcomes and performance |
| Coursework | Academic work assigned to students as part of a course |
| Evaluation Comments | Feedback provided on student performance and areas of improvement |
| Educational Consulting | Professional guidance on educational strategies and systems |
| Admissions Promotion | Strategies and activities to encourage enrollment in educational institutions |
| *Advertising & Marketing* | |
| Sales Letter | Persuasive written communication intended to motivate potential buyers |
| Product Description | Detailed overview of a product's features, benefits, and uses |
| Social Media Content | Engaging text, images, or videos crafted for online platforms |
| Multimedia Script | Planned screenplay integrating various forms of media for marketing |
| Promotional Copy | Compelling text written to boost interest and sales of products |
| Promotional Voiceover | Recorded narration to accompany marketing visuals or ads |
| Travel Guide | Informative content offering insights and tips for travelers |
| Brand Story | Narrative that outlines the history, values, and mission of a brand |
| Personal Blog | Individual commentary or stories shared in an informal online format |
| Marketing Commentary | Analytical thoughts on marketing trends and strategies |
| Slogans | Catchy and memorable phrases designed to convey brand identity |

# C Prompts

## C.1 Query Classification Prompt

Introduced in Appendix A.1.

---
**Classification System Prompt**

You are an expert in query analysis.

---

During each iteration, Put current secondary domain tags under ** Domains **. For example, "1. Academic & Engineering" represents a primary domain, while "1a. Thesis Outline" represents a subdomain.

---
**Query Classification Prompt**

Please determine which of the following domains the query belongs to and identify any stylistic, formatting, or length requirements.

**Example of Format Requirement**: Mimicking the format of uploaded documents, adhering to a given outline format, conforming to academic paper formatting standards, etc.
**Example of Style Requirement**: Suitable for children's reading, rigorous language, humorous tone, etc.
**Example of Length Requirement**: Word count, duration, or other constraints related to output size.

** Query **
{query}

** Domains **
1. Academic & Engineering
    1a. Thesis Outline
    1b. Literature Review

    ...
    1r. Others related to Academic & Engineering

2. Finance & Business
    2a. Market Research
    2b. Sales Report

    ...
    2l. Others related to Finance & Business
...
7. Other
    7a. Others

** Output format **
Return in JSON, strictly output according to the following format, do not output other content
{
    "domain": ["xx.yyy",...]  // Domains involved, such as "6c. Marketing Letter", "8d. Educational Consulting", if the data is invalid and cannot be determined, return an empty list.
    ""style": "", // style requirement if present (e.g. "academic format"), else empty string
    ""format": "", // format specification if present (e.g. "child-friendly tone"), else empty string
    ""length": "" // length constraint (e.g. "500 words") or empty string
}

---

## C.2 Initial Query Generation Prompt

Introduced in Section 3.1.1. You can specify the language of the generated query in the prompt.

---

**Query Classification Prompt**

Generate {NUM} different writing requests under {subdomain} within the context of {primary_domain} in English / Chinese. Ensure the requests are as detailed and specific as possible, and reflect realistic user tone and needs.

Please return in the following JSON format, and do not include anything outside of JSON:
[
   "Writing request 1",
   "Writing request 2",
   . . .
]

---

## C.3 Guidance Pool

Introduced in Section 3.1.1. Randomly select 0-6 items each time.

---

**Query Refinement Guidance Pool**

[
   "Add a requirement for generating specific lengths.",
   "Include format adherence requirements, such as writing according to a prescribed outline or outputting in a specific format.",
   "Add style requirements, like drafting a speech suitable for a particular occasion or adopting the style suitable for a specific audience or mimicking a particular tone.",
   "Incorporate user personalization needs, such as considering the user's identity or integrating personal experiences.",
   "Include more specific content requirements, like details about a particular event or focusing on specific content.",
   "Express concisely in one sentence."
]

---

## C.4 Query Refine Prompt

Introduced in Section 3.1.1.

---

**Query Refinement Prompt**

Please refine and enhance the original writing requirements in the context of generating content in {domain2} from {domain1} based on the provided guidance. Include as many details as possible and indicate whether additional writing materials are needed.

** Original Writing Requirements **
{query}

** Guidance for Modification **
{guidance}

** Output Requirements **
Return the result strictly in the following JSON format, with no additional content outside the JSON:

---

```
{
    "query": "Modified writing requirements",
    "material": "Whether additional reference materials are needed to supplement the writing
requirements. If needed, provide suggestions for the materials; if not needed, return"
}
```

## C.5 Criteria Generation Prompt

Introduced in Section 3.2.

---

**Evaluation System Prompt**

You are an expert evaluator with extensive experience in evaluating the response of a given query.

---

**Criteria Generation Prompt**

Please generate five strict evaluation criteria for assessing the response given the following query. Each criterion should include the following fields: name, criteria_description, 1-2, 3-4, 5-6, 7-8, 9-10.
The criteria should be designed to emphasize detailed assessment and distinguish subtle differences in quality. Ensure that the criteria can discern issues such as relevance, coherence, depth, specificity, and adherence to the query context.
Do not include any additional text. Only output the criteria in the specified JSON format.

\*\* Query \*\*
{query}

\*\* Output format \*\*
```
[
    {
        "name": "first_criteria_name",
        "criteria_description": "Description for the first criteria, emphasizing detailed and
critical assessment.",
        "1-2": "Low score description: Critical deficiencies and major issues that prevent
adequate functionality.",
        "3-4": "Below average score description: Lacking with noticeable shortcomings that
impact overall effectiveness and require improvement.",
        "5-6": "Average score description: Adequate but not exemplary, Baseline performance
that meets essential requirements. Most models may achieve this score.",
        "7-8": "Above average score description: Strong performance characterized by
competent execution, though minor refinements are needed to achieve excellence.",
        "9-10": "High score description: Exceptional performance with all aspects optimally
addressed, demonstrating superior effectiveness and quality without any flaws."
    },
    ...
]
```

## C.6 Rubric-based Scoring Prompt

Introduced in Section 3.2.

---
**Evaluation System Prompt**

You are an expert evaluator with extensive experience in evaluating the response of a given query.

---

Since the query and response may be very long, the {criteria} will appear twice: once before and once after the query and response.

---
**Scoring Prompt**

Evaluate the Response based on the Query and Criteria provided following the Scoring Rules.

** Scoring Rules **
"1-2": "Low score description: Critical deficiencies and major issues that prevent adequate functionality."
"3-4": "Below average score description: Lacking with noticeable shortcomings that impact overall effectiveness and require improvement."
"5-6": "Average score description: Adequate but not exemplary, Baseline performance that meets essential requirements. Most models may achieve this score."
"7-8": "Above average score description: Strong performance characterized by competent execution, though minor refinements are needed to achieve excellence."
"9-10": "High score description: Exceptional performance with all aspects optimally addressed, demonstrating superior effectiveness and quality without any flaws."

-Provide reasons for each score by indicating specific strengths or deficiencies within the Response. Reference exact text passages to justify the score, ensuring that each reason is concrete and aligns with the criteria requirements while highlighting key gaps from the ideal answer.

-Be very STRICT and do not be misled by format or length; ensure that the Response is thoroughly evaluated beyond superficial appearances.

-Carefully discern whether the content of the Response is an illusion, appearing substantial but actually entirely fabricated.

-Sometimes the model may only provide an introduction or an overview without truly completing the query, which should be considered a failed response. Carefully discern this.

-Scoring Range: Assign an integer score between 1 to 10

** Output format **
(Remove symbols that interfere with JSON parsing, don't use " inside reason)
Return the results in the following JSON format, Only output the following JSON format and nothing else:
{
    "score": an integer score between 1 to 10,
    "reason": "Specific and detailed justification for the score using text elements."
}

** Criteria **
{criteria}

---

```
** Query **
{query}

** Response **
{response}

Provide your evaluation based on the criteria restated below:
{criteria}

** Output format **
(Remove symbols that interfere with JSON parsing, don't use " inside reason)
Return the results in the following JSON format, Only output the following JSON format and
nothing else:
{
    "score": an integer score between 1 to 10,
    "reason": "Specific and detailed justification for the score using text elements."
}
```

## D Limitations

This work faces three primary limitations that warrant consideration. First, both the writing model and critic model are primarily trained using conventional SFT approaches, omitting systematic exploration of enhanced optimization strategies. While we demonstrate the partial efficacy of the CoT mechanisms in creative domains, their potential remains unexplored compared to established successes in mathematical reasoning tasks.

Second, our evaluation framework exhibits diminished precision in handling complex multi-dimensional length requirements, including temporal sequencing constraints and section-specific word counts. This limitation underscores the necessity for enhanced scoring methodologies that integrate learned metrics with structured rule-based evaluations to better regulate output specifications.

Third, inherent challenges persist in obtaining reliable pairwise preference annotations for compositional tasks. Despite rigorous annotation protocols, human evaluators inevitably introduce subjective biases when assessing two fair-well responses, particularly regarding narrative preferences and contextual interpretations. While our consensus-building procedures mitigate some variability, absolute alignment with diverse user preferences remains theoretically unattainable.

## E Impact

The introduction of WritingBench and its associated evaluation framework presents significant impacts across multiple dimensions of LLM research and application development:

- **Comprehensive AI Writing Evaluation Ecosystem** WritingBench's extensive query set spans a wide range of domains and requirements, making it an ideal resource for evaluating both general-purpose writing skills and domain-specific expertise. Researchers can use the benchmark to assess a model's overall versatility or focus on specific domains, such as storytelling, scientific writing, or business communication. This dual capability allows for nuanced evaluations that cater to both academic research and practical applications, bridging the gap between theoretical advancements and real-world needs. By publicly releasing WritingBench, including its evaluation protocols, criteria generation tools, and critic models, we contribute to greater transparency and reproducibility in the field of LLM research. Researchers can replicate experiments, validate findings, and build upon our work to further refine evaluation methodologies. Please note that although every released query is manually reviewed to remove harmful content, the responses generated via API calls depend on the underlying model's behavior, cannot be guaranteed to be fully safe or accurate, and should not be used for commercial purposes.

- **Facilitating Domain-Specific Research and Development** With its inclusion of 100 subdomains, WritingBench provides a unique opportunity for domain-specific optimization. For example, indus-

tries such as legal writing, medical documentation, or marketing can evaluate model performance on relevant subsets to select the most suitable models for their needs. Furthermore, WritingBench's query construction strategy can be leveraged to generate diverse queries when specialized evaluation datasets are lacking in a particular field. This capability supports targeted research the development of highly specialized writing tools. We welcome researchers and experts from all fields to join the discussion and collaborate on the development of this benchmark.

- **Advancing Reinforcement Learning and Adaptive Evaluation Frameworks** The query-dependent evaluation framework introduced in WritingBench opens new avenues for integrating adaptive evaluation methods into reinforcement learning pipelines. The ability to generate instance-specific criteria dynamically allows for precise scoring mechanisms that can be used to train models through reward-based optimization. Additionally, the framework's capacity to produce pairwise preference data by comparing predicts against adaptive criteria for improving model alignment with human preferences. This approach enhances the quality of generated text and accelerates the development of models capable of handling complex, multi-dimensional writing tasks.

