# OpenReview forum: "WritingBench: A Comprehensive Benchmark for Generative Writing"
_NeurIPS.cc/2025/Datasets_and_Benchmarks_Track — NeurIPS 2025 Datasets and Benchmarks Track poster_

### Official Review · Reviewer_ozJ4 · 2025-07-01

**Rating:** 5
**Confidence:** 4

**Summary:**

This paper introduces WritingBench, a novel benchmark designed to assess the generative writing capabilities of LLMs. The benchmark comprises 1,000 queries spanning 6 primary domains and 100 subdomains, with explicit evaluation dimensions for style, format, and length requirements. WritingBench proposes a query-dependent evaluation framework where LLMs dynamically generate instance-specific criteria for each writing task, complemented by a fine-tuned critic model for rubric-based scoring. Extensive experiments on 17 LLMs demonstrate the framework's effectiveness and show utility in data curation, where a fine-tuned 7B model using filtered data outperforms GPT-4o in writing tasks.

**Dataset Code Accessibility:**

Yes

**Dataset Code Comments:**

The authors have provided excellent and comprehensive access to all relevant code, data, and metadata, ensuring high reproducibility. The paper includes direct links to a public GitHub repository containing the full WritingBench dataset, evaluation code, and detailed instructions. Beyond the dataset itself, the authors have also open-sourced their key tools, including the fine-tuned critic model and the writing-enhanced models, with links available in the repository. This is further complemented by a public leaderboard on Hugging Face for transparently tracking model performance. Crucially, the authors provide exceptional documentation within the paper's appendix, which details the exact prompts used for query generation, evaluation, and scoring, a critical step for ensuring others can replicate their methodology. This thorough commitment to open-sourcing all assets and providing clear documentation is a major strength of the work.

**Ethical Comments:**

The authors have been diligent and transparent in addressing the ethical considerations of their work, and no significant ethical concerns remain. Their responsible approach is demonstrated by their data handling procedures, which involved using anonymized user data under strict security protocols and implementing a multi-stage review to ensure the final data is free from harmful content. Furthermore, they show exemplary treatment of human subjects, detailing fair compensation ($18 per hour) for annotators and confirming that informed consent was obtained. The authors also proactively considered the societal context of their work through dedicated appendices for both Limitations (Appendix D) and Impact (Appendix E) . Given this comprehensive and responsible handling of key ethical issues, the work does not warrant additional ethical review.

**Ethical Considerations:**

No, there are no or only very minor ethics concerns

**Final Justification:**

The authors' reply is convincing, and I will maintain my rating.

**Limitations Weaknesses:**

- To further bolster the validity of the proposed benchmark, we suggest a more direct validation study. The current pairwise preference comparison is insightful, but having human experts use the same criteria to assign numerical scores would be a valuable addition. Directly comparing these human-assigned scores with the critic model's scores would provide stronger, more granular evidence that the benchmark's scoring mechanism is truly aligned with human expert judgment.
- Bad case analysis is limited as it focuses primarily on length constraints. It is recommended that the authors expand this section to provide a more comprehensive view of model weaknesses. The analysis would be strengthened by including failure cases from other core dimensions, such as models failing to adhere to specific stylistic requirements, incorrectly integrating or contradicting facts from the provided source materials, or struggling with complex formatting specifications.

**Strengths Contributions:**

- Unlike existing benchmarks that often focus on single domains like fiction or use simplistic, templated queries, WritingBench offers unprecedented diversity with 1,000 free-form queries across 100 subdomains. The explicit inclusion of requirements for style, format, and length effectively captures the complexity of real-world writing scenarios.
- This paper proposes a novel query-dependent evaluation framework that moves beyond the limitations of static, predefined assessment criteria. This framework dynamically generates instance-specific criteria and detailed scoring rubrics for each query, which allows for a more nuanced, context-sensitive, and robust evaluation of writing quality.
- The paper validates the value of its benchmark and framework through a rigorous experimental design. It presents a comprehensive benchmark evaluation on a broad set of 17 mainstream models, analyzing their performance across various domains and requirements, while also performing targeted ablation studies to verify the positive impact of CoT on enhancing creative writing capabilities.

---

> ### Author Rebuttal · Authors · 2025-07-31
>
> We sincerely appreciate your thorough review for the encouraging assessment and for the concrete suggestions on strengthening WritingBench. Below we address the main concerns:
>
> ---
>
> ### **1. Human Evaluation Protocol** (Limitation #1)
> While direct numerical scoring using identical rubrics offers intuitive appeal, our pilot studies reveal it could introduce considerable **measurement noise and yields lower reliability**. This stems from two fundamental limitations:
>
> - **Annotator-Specific Biases**: In an early pilot, we asked annotators to rate responses on a 5-point scale and observed pronounced rater bias. Some evaluators were extremely strict on specific dimensions (e.g., formatting compliance or length constraints), while others demonstrated disproportionate leniency. These biases caused substantial shifts in both mean scores and variances across annotators, which would likely be **amplified on a finer 10-point scale**. Given our goal of establishing **stable, human-aligned model rankings**, such scale-drift is problematic. Blind pairwise preference evaluation (Section A.3) achieves higher inter-annotator consistency and better isolates writing quality assessment from subjective scale interpretation.
>
> - **Human Prioritization of Evaluation Criteria**: Annotators also reported that the **relative importance of criteria varies by query**. A technical report, for example, weighs format adherence far more heavily than a lyrical poem. Assigning equal weight to all dimensions through simple averaging dilutes authentic human preferences and amplifies individual biases. To address this, we plan to adopt **weighted aggregation**. The experimental results are shown below:
>
> | Aggregation Method | Alignment Score |
> |--------------------|:------------:|
> | Simple Average     | 84%       |
> | Weighted Average   | 87%       |
>
> The weighted approach better captures human preference patterns and increases the correlation with human judgments. This enhancement will be implemented in our next periodic leaderboard update.
>
> ---
>
> ### **2. Expand Failure Case Analysis** (Limitation #2)
> We will expand more bad case analysis in Appendix A.5 beyond length constraints, to further supplement Section 4.2 (Comparison between LLMs). Some key dimensions failure examples are as follows (due to the space constraints, the original query and response have been shortened here):
> **A. Format Compliance Failure**:
>
> 💡 **Query**
> ```
> Following the given video script format and our school's admission promotion materials, please help me write a video script for campus culture promotion, about 5 minutes long, emphasizing our school's advantageous disciplines, and introducing the application process at the end of the video.
>
> Shooting Script:
> Video Title:   Director:    Shooting Date: 20XX-X-XX
> | Shot | Scene | Script / Subtitle | Music | Duration | Shooting Method |
> |------|-------|-------------------|-------|----------|-----------------|
> | 1    | Opening: Students entering campus with suitcases through the school gate, reporting with admission letters. | Narration: New students bring vitality to this school. | Encounter | 15 s | Close-up of suitcases and admission letters, from far to near |
> | 2    | ... | ... | ... | ... | ... |
>
> College Introduction:
> The Open University of China is directly under the Ministry of Education of the People's Republic of China. It is a new type of higher education institution that implements remote open education, supported by modern information technology, offering both degree and non-degree education.
> ...
> ```
> 📝 **Response** generated by Qwen-Max
> ```
> Video Script for Campus Culture Promotion
> Video Title: Discover Your Future at The Open University of China
> Director: [Your Name]
> Shooting Date: 20XX-X-XX
> Shot 1: Opening Scene – A Glimpse of Our Campus
> Scene: Wide aerial shot of the university campus, transitioning to students walking through the gates with excitement and purpose.
> Script/Subtitle: Narration: "Welcome to The Open University of China, where dreams take flight and futures are shaped."
> Music: Inspiring orchestral piece (e.g., "Awakening" by Two Steps From Hell).
> Duration: 20s
> Shooting Method: Drone footage of campus, slow zoom into students entering the main gate.
>
> Shot 2: Academic Excellence – The Heart of Learning
> ...
> ```
> ⚠️ It failed to follow the specified script-table format and instead produced a section-by-section, enumerated output.
>
> ---
>
> **B. Incorrect Factual Integration**:
>
> 💡 **Query**
> ```
> Please write an analysis report on the revenue of wealth management products for China Merchants Bank and China Construction Bank (with a length of 3000-4000 words), based on the following annual report data of wealth management products from 2018-2022 for both banks:
>
> China Merchants Bank:
> ...
> 2019: ... Total assets: 8.061 billion yuan...
> ...
>
> China Construction Bank:
> ...
> ```
> 📝 **Response** generated by DeepSeek-R1
> ```
> Analysis Report on the Revenue of Wealth Management Products for China Merchants Bank and China Construction Bank (2018–2022)
> ...
> 3. China Merchants Bank (CMB) Wealth Management Analysis
> 3.1 Early Stage (2018–2019)
> ...
> 2019: ...Total Assets: RMB 8.06 billion (wealth management segment)...
>
> ...
>
> 3.3 Key Observations
> Profitability: ...Net profit margin improved from 42.7% (2020) to 62.1% (2022), highlighting operational efficiency...
> ...
> ```
> ⚠️ The source data merely stated “Total assets: 8.061 billion yuan,” with no indication that the amount referred solely to the wealth-management business. The model’s insertion is therefore baseless. Moreover, the claim that the net profit margin “improved from 42.7 % (2020) to 62.1 % (2022)” is entirely fabricated.
>
> ---
>
> Thank you again for your constructive feedback and particularly grateful for the recognition of our benchmark's diversity, dynamic evaluation framework, and commitment to open science. We aim to make WritingBench a **reliable, comprehensive, and long-term benchmark** for generative writing. We believe the planned additions will further advance this goal, and we welcome any additional discussion.

---

> > ### Comment · Reviewer_ozJ4 · 2025-08-05
> >
> > Thank you for the response. I have decided to keep the score unchanged.

---

> > > ### Author Response · Authors · 2025-08-06
> > >
> > > We sincerely appreciate your thoughtful engagement with our rebuttal. We fully respect your decision and value the valuable insights and constructive feedback.

---

### Official Review · Reviewer_Qhwt · 2025-07-02

**Ethics Flags:** Discrimination, bias, and fairness
**Rating:** 4
**Confidence:** 4

**Summary:**

The paper introduces WritingBench, a novel benchmark designed to evaluate large language models (LLMs) across diverse writing tasks, covering 6 primary domains and 100 subdomains. The benchmark emphasizes style, format, and length requirements and proposes a query-dependent evaluation framework where LLMs dynamically generate instance-specific criteria, scored by a fine-tuned critic model. They prove that Smaller models (e.g., Qwen-2.5-7B-filtered) can outperform larger ones (e.g., GPT-4o) in writing tasks

**Dataset Code Accessibility:**

Yes

**Ethical Comments:**

While the paper mentions human annotator compensation, it does not discuss potential biases in query generation (e.g., overrepresentation of Western business/legal norms) or misuse risks (e.g., generating deceptive marketing copy). A broader impact analysis (cf. Appendix E) should address these explicitly.

**Ethical Considerations:**

Yes, there are ethics concerns that require attention by the authors

**Final Justification:**

Thanks for the response. I decide to increase the rating.

**Limitations Weaknesses:**

1. While the benchmark claims real-world relevance, the queries are synthetically generated by Claude 3.7. Although human-refined, their fidelity to organic user requests (e.g., non-expert phrasing, ambiguous requirements) remains unclear. A comparative analysis with real-world writing platforms (e.g., user interaction logs from Grammarly or Medium) could strengthen validity. For instance, given a human-written blog or article on Medium, one could: (1) extract the original writing prompt/objective, (2) generate comparative content via LLMs, and (3) evaluate alignment between LLM outputs and human-written text.
2. The critic model is fine-tuned on Claude-3.7-scored data, raising concerns about generalizability to other evaluators. For example:
    - Would a GPT-4o-trained critic yield comparable results?
    - The paper does not ablate the choice of scorer LLM or explore cross-model evaluation consistency.
3. WritingBench evaluates textual adherence (e.g., style, length constraints) but overlooks:
    - Multimodal writing (e.g., integration of figures/tables in technical reports).
    - Interactive editing (e.g., iterative refinement via human feedback loops).
4. The benchmark highlights model struggles with output length constraints (e.g., outputs often capped at ~3K tokens), but the analysis lacks mechanistic depth. For example:
    - Does performance degradation stem from context truncation, reasoning fragmentation, or repetition?
    - The paper notes that Qwen-Max achieves better performance with longer outputs, while Qwen2.5 Instruct 72B performs worse. This discrepancy warrants investigation into model-specific architectural or training differences (e.g., attention mechanisms, tokenization strategies).

**Strengths Contributions:**

- WritingBench addresses a critical gap by covering 100 subdomains which is comprehensive
- The query-dependent criteria generation (using Claude-3.7) and critic model (fine-tuned on 155K scored samples) achieve 84% human alignment, outperforming static-criteria baselines (67–79%).

---

> ### Author Rebuttal · Authors · 2025-07-31
>
> We sincerely appreciate valuable feedback. Below, we address the key concerns:
>
> ---
>
> ### **1. Benchmark Query Construction Realism** (Limitation #1)
>
> Regarding query authenticity while complying with platform policies and user-privacy regulations, we design a **four-stage human-AI collaborative construction pipeline** (Section 3.1, Figure 2), which implements several strategies for real-world alignment:
> - **Domain Distribution**: To reflect the domain distribution of real-world scenarios, we apply an iterative domain taxonomy construction pipeline, validated against real-world distributions under strict data-security protocols (Appendix A.1).
> - **Requirement Diversity**: Grounded in the heterogeneous needs of users, we build a Query Refinement Guidance Pool that injects diversification strategies for style, format, length, tone, etc (Phase 2 in Section 3.1.1), with statistical validation (Appendix A.1).
> - **Phrasing Authenticity**: To enhance linguistic realism, we first employ multiple LLMs for initial query generation, then apply the expression-related diversification guidance to mimic diverse real-user phrasings, and finally perform expert review and correction in the Expert Optimization stage to ensure queries are harmless and authentic.
>
> ---
>
> ### **2. Evaluator Selection** (Limitation #2)
>
> Our goal is to build an evaluator with **high human-preference alignment** (Section 3.2 details the query-dependent evaluation framework, and Section 4.3 reports its human-consistency experiments). Additional model-selection details and experiments are provided below.
>
> First, we conducted pilot studies on small samples by manually auditing scores and justifications from several SOTA LLMs. After excluding underperforming models, we carried out comprehensive experimental validation on two candidate models (Claude-3.7-Sonnet and GPT-4o) within our proposed query-dependent evaluation framework.
>
> | Evaluator                     | Alignment Score |
> |---------------------------------|:-------------------:|
> | GPT-4o                    | 79%            |
> | Critic model fine-tuned on GPT-4o scores   | 75%            |
> | Claude-3.7-Sonnet       | 87%            |
> | Critic model fine-tuned on Claude scores | 84%        |
>
> The two critic models show a 76% agreement with each other. Overall, Claude-3.7-Sonnet shows superior alignment with human judgments, and the critic model fine-tuned on its scores likewise achieves the highest consistency. Detailed comparison and case analysis are provided in Appendix A.5.2.
>
> ---
>
> ### **3. Scope Clarification and Ethical Concerns** (Limitation #3, Ethical Comments)
> Our benchmark intentionally focuses on assessing the **text-only writing** capabilities of LLMs. The suggested multimodal layouts and iteractive loops would rely on additional components, like front-end renderers, agent pipelines, and UI design, which extend beyond the scope of the present study.
>
> Regarding the mentioned ethical concerns about potential bias in query distribution and the risk of commercial misuse of generated responses, we have taken several precautions:
> - As detailed in the above "Benchmark Query Construction Realism" section, the benchmark queries were crafted to mirror real-world distributions, encompassing **100 subdomains and vary in each domain**. Every query was manually screened to ensure that no harmful content was introduced.
> - Furthermore, the model-generated responses are **used exclusively for offline benchmarking**. They are not shown to end users or employed for marketing.
>
> These safeguards collectively mitigate ethical risks related to query bias and content misuse.
>
> ---
>
> ### **4. Length Constraint Analysis** (Limitation #4)
>
> Our analysis of length constraint adherence is comprehensively addressed through:
> - **In-depth experimental** analysis in Section 4.2 (Key Insights from Requirement Scores).
> - **Supplementary ablation** studies on input/output length factors in Appendix A.4.
> - **Case studies** of typical length compliance failures (e.g., violations of sentence-level length constraints) like in Appendix 7.
>
> Regarding the suggested investigation into model-specific architectural or training differences between Qwen-Max and Qwen2.5-72B, such comparative analysis is unfortunately infeasible. Qwen-Max is closed-source, and its architecture, tokenizer, and training procedure have not been disclosed.
>
> ---
>
> We sincerely value your feedback and are pleased to receive positive comments on the **comprehensive converage of our benchmark with real-world scenarios** and the recognition of our **query-dependent evaluation framework**. Thank you for considering our responses. We look forward to your responses and further discussions on this topic.

---

> ### Author Response · Authors · 2025-08-05
>
> Dear Reviewer,
>
> Thank you again for your feedback. As the discussion period is coming to a close, we would greatly appreciate it if you could let us know whether our responses have adequately addressed your concerns. In particular, the ethical comments you raised have been fully discussed in our latest Ethics Review submission. We are eager to resolve any remaining questions or comments you might have.
>
> Please let us know if there’s anything further we can provide or discuss. We are thankful for your time and consideration and look forward to your response.
>
> Best regards,
>
> The Authors

---

> ### Author Response · Authors · 2025-08-08
>
> Dear Reviewer,
>
> I hope this message finds you well. As the rebuttal deadline is quickly approaching, we wanted to kindly follow up and inquire whether my previous responses have sufficiently addressed your comments and concerns. If there is any aspect still unclear, or if you'd like more clarification or discussion regarding our work, please feel free to let us know at any time.
>
> Thank you very much for your attention and consideration. We look forward to your reply.
>
> Best regards,
>
> The Authors

---

### Official Review · Reviewer_tHAn · 2025-07-02

**Rating:** 5
**Confidence:** 3

**Summary:**

This paper proposes WritingBench, a benchmark for generative writing. 6 core writing domains and 100 subdomains are included. In addition, a fine-tuned 7B critic model could score on instance-specific criteria, which are generated by a query-dependent evaluation framework.

**Dataset Code Accessibility:**

Yes

**Dataset Code Comments:**

code is provided in the supplementary material.

**Ethical Considerations:**

No, there are no or only very minor ethics concerns

**Final Justification:**

Authors' further clarification addressed my concerns the saturation. I believe that continuously maintaining and updating this benchmark is important.

**Limitations Weaknesses:**

- According to Table 3, one concern is whether this benchmark will remain useful and challenging in the near future. The overall scores of *-filtered 7/8B models are catching up with pioneer models such as Claude-3.7 and Deepseek-R1. Is there a saturation at which models are squeezed into a metric space with only marginal distinctions between them?

**Strengths Contributions:**

- This paper is easy to read and understand.
- The work is complete, covering query&criteria synthesis and scoring. Multi-step synthesis and human annotation improve both diversity and quality. The proposed instance-specific criteria dynamically adapt to the query.
- The experiments are comprehensive, including baselines on many popular models, human consistency, and ablations on data curation & length.

---

> ### Author Rebuttal · Authors · 2025-07-31
>
> We sincerely appreciate your insightful concern regarding benchmark longevity and saturation risks. WritingBench is designed to serve as a **reliable, open, sustainable benchmark** for evaluating LLM's writing capabilities. Below we clarify how we deal with this concern and how the benchmark will remain challenging and informative in the long run.
>
> ---
>
> ### **1. Periodic Version Updates to Maintain Discriminative Power** (Limitation #1)
> Given the rapid advances in LLM writing quality over the past six months, we see **periodic upgrades as essential to the benchmark’s lifecycle**. Since the initial release earlier this year, we have already rolled out a major version update to keep the tasks challenging and prevent score saturation. Key enhancements include:
> - **Discriminative Query Library**: We removed oversimplified prompts (i.e., those on which most models scored above 8) and added new queries that probe emerging, more demanding writing challenges.
> - **Finer-Grained Criteria and Rubrics**: The query-aware criteria and rubrics were further refined through improved dynamic-criteria generation, preventing the evaluator from awarding high scores too easily and producing more discriminative results.
> - **More Accurate Evaluator**: We migrate to Claude-3.7-Sonnet for training the critic model and scaled the training data from 50K to 155K spanning approximately 40 models. This upgrade increased critic model's human alignment score from 82.7% to 84.3% and improved its generalisation.
>
> ---
>
> ### **2. Quantitative Evaluation of the prerodic Upgrade**
> Below we provide a comparison of evaluation scores between the new and old leaderboards:
> | Models                    | last version | current version |
> |-------------------|:----------:|:---------:|
> **Proprietary LLMs** | | |
> Claude-3.7-thinking  |  8.72  | 7.91 |
> Claude-3.7                | 8.68  | 7.85 |
> Qwen-Max                | 8.37 | 7.16 |
> o1-Preview                | 8.15 | 6.89 |
> GPT-4o                     | 8.16 |  6.81 |
> Gemini-1.5-Pro        | 7.78 | 6.21 |
> **Open-source LLMs** | | |
> Deepseek-R1         | 8.55 | 7.70 |
> Deepseek-V3         | 7.95 | 6.35 |
> Mistral-Large-Instruct   | 7.64 | 6.00 |
> Qwen-2.5-72B-Instruct | 7.90 | 6.40 |
> Qwen-2.5-7B-Instruct   | 7.43 | 5.64 |
> Llama-3.3-70B-Instruct | 7.01 | 5.05 |
> Llama-3.1-8B-Instruct   | 6.35 | 4.42 |
> **Capability-enhanced LLMs** | | |
> Suri                                | 4.97 | 3.21 |
> LongWriter                     | 7.91 | 6.27 |
> Qwen-2.5-7B-filtered    | 8.49 | 7.44 |
> Llama-3.1-8B-filtered    | 8.49 | 7.39 |
>
> Notably, the score compression at the top end has been substantially alleviated. The new version restores clear separation between models and leaves **fresh headroom for future progress**. Like other continuously evolving benchmarks such as LiveBench, we actively **track new model releases and carry out periodic updates** to the query pool, rubrics, and evaluator to support the community in developing ever-stronger writing models.
>
> ---
>
> We sincerely value your feedback and deeply appreciate your recognition of WritingBench’s comprehensive design, rigorous evaluation, and experimental validity. Our goal is to make WritingBench a **sustainable community resource for charting the frontiers of generative writing**. We hope these clarifications address your concerns and we look forward to further discussion.

---

> > ### Comment · Reviewer_tHAn · 2025-08-05
> >
> > Thank you for your further clarification, which has addressed my concerns. I believe that continuously maintaining and updating this benchmark is important, and I hope this aspect can be added to the paper. I will raise my score accordingly.

---

> > > ### Author Response · Authors · 2025-08-06
> > >
> > > Thank you for your positive feedback and for raising the score. We appreciate your emphasis on the importance of continuous benchmark maintenance and will explicitly add this to the paper.

---

> ### Author Response · Authors · 2025-08-05
>
> Dear Reviewer,
>
> Thank you again for your feedback. As the discussion period is coming to a close, we would greatly appreciate it if you could let us know whether our responses have adequately addressed your concerns. Please let us know if there’s anything further we can provide or discuss. We are thankful for your time and consideration and look forward to your response.
>
> Best regards,
>
> The Authors

---

### Official Review · Reviewer_yfnY · 2025-07-02

**Rating:** 4
**Confidence:** 3

**Summary:**

WritingBench introduces a comprehensive benchmark for evaluating large language models in generative writing across 6 primary domains and 100 subdomains. It features a novel query-dependent evaluation framework that dynamically generates instance-specific criteria, enabling nuanced and context-aware assessment with strong human alignment. Extensive experiments demonstrate the effectiveness of the framework in benchmarking, data curation, and training smaller models that outperform larger counterparts on writing tasks.

**Dataset Code Accessibility:**

Yes

**Ethical Considerations:**

No, there are no or only very minor ethics concerns

**Limitations Weaknesses:**

-  The authors didn't conduct ablation studies on additional post-training methods, such as Reinforcement Learning with Human Feedback (RLHF) and Direct Preference Optimization (DPO), to further validate the performance.
- The pipeline heavily depends on predefined heuristics and specific LLMs (Claude-3.7, GPT-4o) for generation, which can significantly introduce bias.
- as a dataset benchmark reviewer, I do care more about the codebase and dataset document's clarity, it seems the codebase and dataset documents is not fully polished, without enough in-depth documenting, how can they guarantee the follow-up researchers can quickly ramp up on the research of this direction?

**Strengths Contributions:**

- The authors open-sourced not only the WritingBench dataset, but also the evaluation protocols, criteria generation prompts, and critic model
- The benchmark includes 555 English and 445 Chinese queries, enabling cross-lingual evaluation and showcasing how LLMs perform across languages, which is important for global applicability.

---

> ### Author Rebuttal · Authors · 2025-07-31
>
> We sincerely appreciate the constructive feedback and positive assessment of WritingBench's contributions. Below we address the main concerns:
>
> ---
>
> ### **1. Ablation Studies of Post-training** (Limitation #1)
> The **primary focus** of this work is to establish a **comprehensive benchmark for evaluating LLM's writing capabilities**. The two small-scale SFT writing-oriented models in Section 3.3 (Evaluation-Guided Data Curation for Writing Enhancement) serve solely to validate our proposed query-dependent evaluation framework's ability to identify high-quality writing responses, as experimentally confirmed in Section 4.4.
>
> Evidence that post-training methods can further enhance writing performance is explored in our concurrent work [1]. Key results across three writing benchmarks are shown below:
>
> | Model | SFT | RL | WritingBench | EQ-Bench | LongBench-Write |
> |--------------|:-------------:|:---------------------------:|:--------------:|:---------:|:-----------------:|
> | Qwen2.5-7B-Instruct       | ✗       |      ✗             | 5.64        | 49.59   | 8.50           |
> | Qwen2.5-7B-SFT            | ✓   | ✗               | 7.44        | 70.02   | 9.22           |
> | **Qwen2.5-7B-RL**       | ✓ |  PPO                   | **7.91**        | **73.19**   | **9.31**           |
> | Llama3.1-8B-Instruct      | ✗      |      ✗                     | 4.42        | 48.40   | 7.39           |
> | Llama3.1-8B-SFT           | ✓    | ✗                 | 7.39        | 78.11   | 9.07           |
> | **Llama3.1-8B-RL**      | ✓ | PPO                   | **7.85**        | **82.73**   | **9.24**           |
>
> These results illustrate that once the data-curation with WritingBench is applied for data filtering (SFT), additional post-training methods could yields further gains. While we welcome further discussion of future enhancement strategies, such explorations extend beyond the scope of the present paper.
>
> ---
>
> ### **2. Mitigating Possible Bias from LLMs** (Limitation #2)
> Human-AI collaboration brings advantages in efficiency and scalability. To mitigate potential bias in **query generation** and **evaluation**, we implemented rigorous safeguards:
>
> First, we propose a four-stage human–AI collaborative pipeline for query construction (Section 3.1, Figure 2). To minimize bias and reflect real-world domain distributions, we employ the following strategies:
>
> - **Wide Domain Coverage and Diverse Requirement Dimension**: We develop an iterative domain-taxonomy pipeline and a set of query refinement strategies (Phase 2 in Section 3.1.1), both grounded in heterogeneous user needs. These components are validated against real-world distributions under strict data-security protocols (Appendix A.1). The process ultimately produces 1,000 queries spanning 100 subdomains with substantial in-domain variation.
> - **Linguistic Authenticity and Content Auditing**: In the final-stage expert optimization, we implement manual correction of all queriesto guarantee authentic, unbiased, and harmless content.
>
> For evaluation, we adopt human-preference consistency as our core validation metric. Our goal is to select a **evaluator that achieve high human preference alignment**. After manually auditing the scores and justifications produced by several SOTA LLMs in a pilot study, we conducted comprehensive tests on the two most promising candidates: Claude-3.7-Sonnet and GPT-4o. Experiments in Section 4.3 shows that the dynamic, query-aware metrics in our framework achieve higher agreement with human judgments than static metrics. We further assessed the two candidate LLMs and the critic models fine-tuned on their scores:
>
> | Evaluator                     | Alignment Score |
> |---------------------------------|:-------------------:|
> | GPT-4o                    | 79%            |
> | Critic model fine-tuned on GPT-4o scores   | 75%            |
> | Claude-3.7-Sonnet       | 87%            |
> | Critic model fine-tuned on Claude scores | 84%        |
>
>
> Claude-3.7-Sonnet demonstrates notably superior alignment, and the critic model distilled from its judgments retains high consistency (additional case studies in Appendix A.5.2). This choice of evaluator is also adopted by EQ-Bench, another benchmark specially for creative-writing, reinforcing its suitability.
>
> ---
>
> ### **3. Codebase & Documentation** (Limitation #3)
> We respectfully confirm that this work adheres to the NeurIPS transparency standards. All materials, including benchamrk dataset, evaluation codes, prompts, instruction documents, critic model, and writing model, are already **publicly avaible** via the Github link listed under the abstract. Recent community adoption includes use in independent research [1-2] and the Qwen3 model release [3].
>
> Leaderboard maintenance and iteration are part of our **ongoing effort toward continuous improvement**. We actively incorporate feedback and reported issues into **regular updates**, and we eagerly welcome specific enhancement suggestions for implementation.
>
> ---
>
> We sincerely appreciate your thoughtful comments and are pleased with the recognition of the breadth, bilingual coverage, and openness of WritingBench. Our goal is to provide a **reliable, open, bilingual benchmark** and a highly human-aligned evaluation framework. The ablations, public code, and ongoing maintenance aim to support the community in developing stronger writing models. We hope these clarifications address your concerns and look forward to further discussion.
>
> ---
>
> [1] Lei X, Li C, Wu Y, et al. Writing-RL: Advancing Long-form Writing via Adaptive Curriculum Reinforcement Learning[J]. arXiv preprint arXiv:2506.05760, 2025.
>
> [2] Lu X. Writing-Zero: Bridge the Gap Between Non-verifiable Problems and Verifiable Rewards[J]. arXiv preprint arXiv:2506.00103, 2025.
>
> [3] Yang A, Li A, Yang B, et al. Qwen3 technical report[J]. arXiv preprint arXiv:2505.09388, 2025.

---

> > ### Comment · Reviewer_yfnY · 2025-08-07
> >
> > Thank you for the detailed and thoughtful rebuttal. I appreciate the authors’ efforts to address the raised concerns, particularly regarding the scope of the benchmark, mitigation of LLM-induced bias, and transparency of the codebase.
> >
> > Overall, I believe the paper makes a solid contribution and has potential for impact in generative writing evaluation.  I’m happy to maintain my current score and support acceptance of the paper.

---

> > > ### Author Response · Authors · 2025-08-08
> > >
> > > Thank you for your valuable feedback and acceptance decision. We sincerely appreciate your contribution to improving this work.

---

> ### Author Response · Authors · 2025-08-05
>
> Dear Reviewer,
>
> Thank you again for your feedback. As the discussion period is coming to a close, we would greatly appreciate it if you could let us know whether our responses have adequately addressed your concerns. Please let us know if there’s anything further we can provide or discuss. We are thankful for your time and consideration and look forward to your response.
>
> Best regards,
>
> The Authors

---

### Note · Authors · 2025-08-13

Dear Area Chair,

We sincerely appreciate the constructive feedback provided by reviewers yfnY, tHAn, Qhwt, ozj4, and ethics reviewers EfZA, QzBr. Below, we summarize key strengths noted by reviewers and outline specific revisions addressing core concerns:

---

### **Strengths Highlighted by Reviewers**
- **Comprehensive Benchmark Design** (yfnY, tHAn, ozJ4, Qhwt): Reviewers acknowledged our innovative human-AI collaboration pipeline, creating 1000 free-form queries across 100 subdomains mirroring real-world diversity.
- **Dynamic Evaluation Framework** (yfnY, tHAn, Qhwt, ozJ4): The query-dependent evaluation framework and its critic model were recognized as significant advances beyond static methods.
- **Rigorous Experimental Validation** (yfnY, tHAn, ozJ4): Extensive experiments and thorough ablation studies confirm the benchmark’s practical utility and support for thorough, multi-dimensional analysis.
- **Community Contribution** (yfnY, ozJ4): We open-sourced the WritingBench dataset, evaluation protocols, and full construction prompts for reproducibility, complemented by a public leaderboard for transparent performance tracking.

---

### **Revisions Made to Address Key Concerns**
- **Enhanced Consistency Validation** (yfnY, Qhwt, ozJ4): To better validate the evaluator with human preferences, we conducted additional consistency comparisons of candidate models, extending the consistency study in Section 4.3.
- **Expanded Failure Case Analysis** (Qhwt, ozJ4): To further supplement the multi-dimension analysis between LLMs in Section 4.2, we broadened writing failure coverage in Appendix A.5 with more detailed bad cases.
- **Strengthened Misuse Safeguards** (Qhwt): We will add explicit prohibitions, permitting only offline benchmarking and forbidding commercial deployment, to prevent misuse.
- **Periodic Benchmark Evolution** (tHAn): We see periodic upgrades as essential to the benchmark’s lifecycle to keep the tasks challenging and informative in the long run.

---

We sincerely appreciate all reviewer comments. Reviewers yfnY, tHAn, and ozj4 confirmed their concerns were addressed and gave positive feedbacks, though we remain unclear whether reviewer Qhwt’s concerns have also been addressed; and we are fully open to any further discussion. Our goal is to make WritingBench a **reliable, comprehensive, sustainable** community resource for charting the frontiers of generative writing. Thank you again for your support.

Best regards,
The Authors

---

### Decision · Program_Chairs · 2025-09-18

**Decision:**

Accept (poster)

**Comment:**

The paper proposes a new benchmark dataset, WritingBench, for generative writing.

Overall, the reviewers recognized the comprehensiveness of the study, for example, the broad scope (6 primary domains and 100 subdomains) and solid evaluation protocols (open criteria generation prompts and critic model).

On the other hand, certain criticisms were raised during the review. For example, Reviewer yfnY asked ablation study on post-training, which was addressed during rebuttal. Reviewer tHAn questioned the future relevance of the dataset due to potential performance satuation. The authors explained that their contribution is not only about the dataset, but also the framework, so they will continuously maintain and upgrade the dataset.

Ethics reviews reveal that the potential bias issue can be discussed in the revision. But this is relatively minor.